# Central terminals of primary afferents coordinate the spontaneous activity of dorsal horn neurons

Javier Lucas-Romero 🆔, Jose Antonio Lopez-Garcia 🆔 and Ivan Rivera-Arconada 🆔

*Department of Systems Biology, University of Alcala, Madrid, Spain*

The peer review history is available in the Supporting Information section of this article (https://doi.org/10.1113/JP287970#support-information-section).

**Abstract figure legend** Primary afferents transmit sensory information to second-order neurons in the spinal cord, but also receive presynaptic contacts regulating synaptic transmission. Opto- and chemogenetic modulation of the excitability of primary afferent terminals reveals its fundamental role in synchronising spontaneous activity of dorsal horn neurons. Spontaneous and evoked activity in the dorsal root was abolished during primary afferent depolarisation produced by channelrhodopsin activation. In contrast, in dorsal horn neurons spontaneous firing was augmented but the occurrence of coordinated activity was impaired. Light-induced hyperpolarisation of primary afferents potentiated the spontaneous and evoked dorsal root potentials and the coordination in the firing of dorsal horn neurons. Activation of the hM4Di receptor expressed in primary afferents inhibited dorsal root responses and synchronic events, but, in contrast to the depolarisation, the activity of dorsal horn neurons was reduced. These results demonstrate the role of primary afferents as a fundamental element in the generation of coordinated activity in spinal circuits. CNO, clozapine N-oxide.

**Javier Lucas-Romero** is a Postdoctoral Research Associate at Washington University in St. Louis. He completed his PhD in Cellular Signalling at the University of Alcalá under the supervision of Dr Jose A. Lopez-Garcia. His doctoral research focused on classifying and elucidating the origin of spontaneous activity patterns in neurons of the superficial dorsal horn of the spinal cord, a key region involved in pain processing. Currently, under the supervision of Dr Jacob G. McPherson, his research explores the role of these spontaneously active neurons in the complex processing of nociceptive information. He is particularly interested in how this activity is altered under pathological conditions such as inflammatory or neuropathic pain and spinal cord injury. Additionally, he is investigating spinal cord stimulation as a potential therapeutic approach for neuropathic pain and spinal cord injury.

The Journal of Physiology

**Abstract** Central terminals of primary afferents and dorsal horn neurons usually exhibit spontaneous activity, the two phenomena being interrelated. Spontaneous activity may constitute a system for adjusting the level of excitation of spinal circuits and the processing of somatosensory information. Superficial dorsal horn neurons fire action potentials in a coordinated form, giving rise to population events. These population events are altered by peripheral inflammation, suggesting their implication in central sensitisation. In this work, we aimed to define the role of primary afferents in the occurrence of this coordinated activity. Channelrhodopsin-2, archaerhodopsin-3 or the hM4Di-DREADD receptor were expressed in primary afferents by Cre-recombination under control of the advillin promoter. Dorsal roots and superficial dorsal horn neurons were simultaneously recorded using *in vitro* spinal cord slices from neonatal mice. Depolarisation of primary afferents by activation of channelrhodopsin-2 inhibited dorsal root activity and the coordinated firing of dorsal horn neurons. DREADD activation reduced the activity in the afferents and depressed coordinated activity in dorsal horn neurons. In contrast, hyperpolarisation of afferents by archaerhodopsin-3 augmented dorsal root responses and increased the coordinated activity of spinal neurons. The present results demonstrate a direct implication of primary afferents in the generation of coordinated spontaneous firing in superficial dorsal horn neurons.

(Received 28 October 2024; accepted after revision 21 May 2025; first published online 7 June 2025)

**Corresponding author** I. Rivera-Arconada: Departamento de Biología de Sistemas, Edificio de Medicina, Campus Externo, 28805 Alcalá de Henares, Madrid, Spain. Email: ivan.rivera@uah.es

**Key points**

- The input of somatosensory information through primary afferents is a process subjected to regulation at the level of the spinal cord, even before it reaches second-order neurons.
- Primary afferent and spinal cord neurons exhibit spontaneous activity, which is altered in pathological models of pain.
- This study demonstrates the role of primary afferents as a fundamental coordinating element for the spontaneous activity of dorsal horn neurons.
- These results show that modulating the activity of the central terminals of primary afferents may have profound implications in both the excitability of spinal cord circuits and the processing of somatosensory information.

# Introduction

The spinal cord is the first relay and processing centre for somatosensory information. Primary afferents send information in an organised manner within the different laminae of the spinal cord, depending on the modality of the information they transmit (Basbaum et al., 2009). However, the ability of spinal circuits to integrate and process sensory information depends critically on their level of activity and excitation. Neurons in the spinal cord with autogenous spontaneous activity are generators that maintain and adjust the level of excitability of local spinal circuits (Inacio et al., 2016; Li & Baccei, 2011; Lucas-Romero et al., 2024; Luz et al., 2014). Spontaneous activity also involves primary afferent terminals. They receive synaptic input from spinal neurons causing depolarisation and firing of action potentials at the central terminals that can be transmitted anti- and

orthodromically (Bos et al., 2011; Lucas-Romero et al., 2022). Spontaneous activity of afferent terminals and spinal neurons is enhanced in models of inflammatory pain, suggesting a relationship to central sensitisation (Lucas-Romero et al., 2022; Vicente-Baz et al., 2022).

Regulation of excitability at central terminals of primary afferents is also critical in determining their ability to transmit information to second-order spinal neurons. The excitability of primary afferents is controlled by presynaptic contacts on the terminals that can adjust the level of neurotransmitter release (Rudomin & Schmidt, 1999). Historically, the key neurotransmitter for these contacts has been considered to be GABA, which acts on GABA-A receptors to induce depolarisation due to an unusually high chloride concentration in primary afferents (Price et al., 2009). Depolarisation, in turn, may reduce neurotransmitter release on arrival of action potentials (Eccles et al., 1961; Mendell & Wall,

1964). However, the inhibitory effect of these presynaptic contacts has been questioned recently (Hari et al., 2022). In other areas of the CNS, changes in membrane potential of axonal terminals have been reported to exert either excitatory or inhibitory actions (Jang et al., 2001, 2002; Khatri et al., 2019; Kramer et al., 2020; Ruiz et al., 2010; Wang et al., 2019; Zbili & Debanne, 2019; Zorrilla de San Martin et al., 2017).

Despite its high relevance to the functioning of the spinal cord, the effects of primary afferent depolarisation and hyperpolarisation on the activity of spinal circuits have not been directly studied. Here we address this issue by using mutant mice expressing rhodopsins under control of the advillin promoter to manipulate the membrane potential of central terminals of primary afferents in a controlled manner. Simultaneous electrophysiological recordings from a dorsal root and dorsal neurons were obtained from an *in vitro* spinal cord preparation. The results obtained show that depolarisation of the afferents by channelrhodopsin-2 abolished dorsal root responses and synchronisation of firing in dorsal horn neurons. Similar effects occurred after inhibition of afferents with DREADDs. In contrast, hyperpolarisation of afferents resulted in augmented dorsal root responses and increased coordinated activity in spinal neurons. These results demonstrate the role of primary afferents in coordinating spontaneous activity and regulating the excitability of spinal circuits.

## Methods

### Animal models and ethical approval

The results presented in this work were obtained from 47 mice bred at the Animal Facility of the University of Alcala, which derived from mutant animal lines obtained from The Jackson Laboratories (Bar Harbor, ME, USA). The animals used expressed channelrhodopsin-2/EYFP (ChR2, strain #024109), hM4Di-DREADD/mCitrine (strain #026219) or archaerhodopsin-3/EGFP (ArchT, strain #021188) following exposure to Cre recombinase enzyme. To obtain their expression in primary afferent neurons, homozygous female mice with these genotypes were mated with heterozygous male mice expressing Cre recombinase under control of the advillin promoter (AdvillinCre knock-in/knock-out mice; strain #032536).

Animals of both sexes, mostly between 8 and 12 days old (4 and 8 g body weight), were used. One 22-day-old animal was used to obtain longer peripheral nerves for the analysis of A- and C-fibre volleys. An additional group of 5-week-old animals from the three phenotypes was used to visualise the expression of fluorescent reporter proteins. Animals were housed in groups with their respective parents and siblings with no limitation for food or water. All experiments were carried out in accordance with

national and EU legislation, and the procedures were approved by the Ethics Committee of the University of Alcala (CEI-UAH-AN 2019/008) and the Community of Madrid (PROEX 51.0/20).

During the procedures, every effort was made to reduce animal suffering and discomfort. To reduce the number of animals required, when possible, two recording sessions were performed per animal. In these cases, a second study was performed using a different dorsal root and multi-electrode position, in a different segment of the spinal cord, and the recording period started at least 1 h after the previous one.

Identification of the animals' genotype was done by PCR after obtaining post-mortem tissue samples from the animals used in electrophysiological experiments. Both the protocol and the primers indicated by the breeder were used. Phenotype was also verified by the expression of fluorescent proteins in the dorsal root ganglion and the spinal cord. In animals with channelrhodopsin-2 expression, successful expression was also confirmed by the withdrawal reflex response elicited *in vivo* in response to 455 nm light stimulation. The investigators could only be blinded to the phenotype of the animals in the DREADD studies, since in the other cases the application of light during recording unveiled the phenotype.

### *In vitro* spinal cord preparation and optogenetic stimulation

The spinal cord was dissected by a dorsal laminectomy under deep anaesthesia with urethane (2 mg/kg), and the mice were killed by cervical dislocation while still under anaesthesia. The meninges were removed into a Petri dish filled with sucrose-substituted ACSF, oxygenated (95% $O_2$, 5% $CO_2$) and ice cooled, and then sectioned with a vibratome in the same medium. A single horizontal slice of 500 μm was obtained, containing the superficial laminae of the dorsal horn with dorsal roots attached. The slice was placed dorsal side down in a Sylgard-bottom organ bath and fixed with pins. The slice was continuously bathed with oxygenated ACSF at room temperature (22 ± 1°C). The composition of the ACSF was (mM): NaCl (127), KCl (1.9), $KH_2PO_4$ (1.5), $MgSO_4$ (1.3), $CaCl_2$ (2), $NaHCO_3$ (22) and glucose (10) at pH 7.4. For the optogenetic experiments, the light source was connected to an optical fibre, which was placed at ∼0.5 cm from the preparation, directly above the bath. With this disposition, light illuminated both the spinal cord and the dorsal roots.

Channelrhodopsin-2 was stimulated with a LED at 455 nm (#M455F3; #LEDD1B controller; Thorlabs, Newton, NJ, USA) connected to a 400 μm optical fibre (#M28L02; Thorlabs). The irradiance at a distance of 0.5 cm, measured with a PM100USB meter and a photodiode power sensor S120VC (Thorlabs), could be adjusted

between 1 and 42 mW/cm$^2$. A similar arrangement with a 565 nm LED (# M565F3; Thorlabs) was tested for archaerhodopsin-3 stimulation, but the responses recorded *in vitro* were very small due to the low irradiance produced (<20 mW/cm$^2$). For this reason, a more powerful system was built using a 567 nm LED (#SP-02-L1; LED driver #A009-D-V-1000 BuckBlock; Luxeon Star LEDs, Lethbridge, Canada) coupled to a 5 mm diameter optical fibre. Using the same measurement system as above, the irradiances used in the *in vitro* studies ranged from 30 to 187 mW/cm$^2$. Light application to the electrode holder was avoided to prevent photoelectric signals that can alter the correct measurements of basal potential.

### Electrical stimulation of dorsal roots and electrophysiological recordings

With the slice in the bath, a dorsal root (L3 or L4) was inserted into a tight-fitting glass suction electrode to electrically activate the primary afferents contained in the root. Stimulus intensity was adjusted to stimulate all types of primary afferents (200 μs and 300 μA; Rivera-Arconada & Lopez-Garcia, 2006).

A second dorsal root (L3 or L4) was inserted into a suction electrode to record the signals from the dorsal root, both spontaneous and those induced by electrical and light stimulation. The signal obtained from the root was fed to an Axoclamp 2B (Molecular Devices, Sunnyvale, CA,, USA), amplified ×100, low pass filtered at 10 kHz (Neurolog System, Digitimer, Fort Lauderdale, FL, USA) and digitised at 20 kHz (RHS2000 Stimulation/Recording Controller, Intan Technologies, USA) for further analysis using Spike2 and Matlab.

Recordings from superficial dorsal horn neurons were obtained with 32–64 channel multi-electrode arrays (MEAs; #M4×8-5mm-Buz-200; Neuronexus, Ann Arbor, MI, USA). The arrays were mounted on a motorised micromanipulator and lowered into the bath chamber until the tip touched the Sylgard bottom. Using this reference, the electrodes were then moved on the top of the slice and inserted into it until the tip was at 50–200 μm from the Sylgard bottom reference. After a long stabilisation period in the absence of any type of stimulation (∼60 min), spontaneous as well as electrical and light-induced responses were recorded. The recorded signals were amplified, filtered between 0.2 and 3 kHz and digitised at 20 kHz using an RHS2000 stimulation/recording controller amplifier (Intan Technologies). These signals were then converted to Spike2 and Matlab compatible formats for analysis. Spike sorting was performed using kilosort2 software (Pachitariu et al., 2024), followed by detailed manual monitoring using Spike2 to refine the automated analysis.

### Peripheral nerve recordings

To further investigate the effects of light stimulation in animals with channelrhodopsin-2 expression, the saphenous and sciatic nerves were extracted from two animals (10 and 22 days old) and maintained *in vitro* with synthetic interstitial fluid (in mM: 108 NaCl, 3.48 KCl, 0.7 MgSO$_4$, 26 NaHCO$_3$, 1.7 NaH$_2$PO$_4$, 1.53 CaCl$_2$, 9.6 sodium gluconate, 5.55 glucose and 7.6 sucrose). The nerves were placed in the same recording chamber used for the spinal cord and glass suction electrodes were placed at both ends for stimulation and recording. Signals were processed with an Axoclamp 2B (Molecular Devices), amplified ×100 and low pass filtered at 10 kHz (Neurolog System, Digitimer) and digitised at 20 kHz using a 1401micro and Spike2 software (CED, Cambridge, UK). The optic fibre was directed to the distal end of the nerve for optogenetic stimulation. The volleys produced by light activation of primary afferents were measured and compared to signals obtained in response to electrical stimulation.

### Histological procedures

To verify the expression of rhodopsins and DREADD in primary afferents, the expression of fluorescent reporter proteins was examined. For this, both dorsal root ganglia and the spinal cord from the neonatal animals used in the electrophysiology experiments, as well as from other adult animals used ad hoc, were extracted and conserved. In all cases it was possible to visualise fluorescence from fresh tissue, both in the ganglion and in the spinal cord. For better visualisation, some samples were fixed by immersion in 2–4% paraformaldehyde for 4–20 h, washed in phosphate buffer and cryoprotected overnight in 30% sucrose. Sections of 40 μm were cut in a cryostat and fluorescence was visualised directly under the microscope. The fluorescent protein mCitrine did not resist the fixation procedures even under the mildest conditions tested and its expression could only be confirmed on unfixed tissue.

### Experimental design

For optogenetic studies, the preparation was kept in the bath with the electrodes in their final position in the absence of stimulation for 60 min. Basal spontaneous activity was then recorded from the dorsal root and dorsal horn neurons for at least 15 min in the absence of stimuli. The protocol then followed with light pulses of 60 s duration and increasing intensity that were applied separated by 120 s. Later, electrical stimuli were applied to the dorsal root in the absence and presence of light. In the archaerhodopsin-3 studies, four different light intensities were used: 30.5, 113.5, 169.4 and 187.2 mW/cm$^2$. For channelrhodopsin-2 experiments, seven intensities

between 1.0 and 42.3 mW/cm$^2$ were applied. For these latter studies short light pulses (2–50 ms) and ramped stimulation protocols with a final intensity of 42.3 mW/cm$^2$ were also used. Ramp stimuli consisted of a linear increase of light intensity during 20 s until reaching maximum intensity, which was then maintained for a further 40 s.

In the hM4Di-DREADD experiments, basal activity was recorded during 25 min and then dorsal root electrical stimuli were applied before the perfusion to the entire slice with clozapine N-oxide (CNO) at 10 µM for 30 min. At the end of the application period, electrical stimuli were repeated. All ACSF components and CNO were purchased from Sigma-Aldrich (Merck, Madrid, Spain).

### Data analysis and statistics

For dorsal root recordings, the frequency and amplitude of spontaneous dorsal root potential (sDRP) and the basal potential were measured. In addition, the integrated area of the evoked dorsal root potential (eDRP) was measured during the first second following the electrical dorsal root stimulation.

For dorsal horn neuron recordings, the activity of each isolated neuron was represented into a single channel after spike sorting. Then the combined firing activity was measured by accumulating in the same channel the firing of all neurons recorded and then representing the data as the mean frequency at 0.2 s intervals. The coordination in spontaneous firing in the sample was analysed as population bursts. These population bursts were detected by a threshold that was manually adjusted above basal firing frequency in control conditions. The ratio between manual threshold and basal firing frequency was calculated and was used to readjust the threshold in the different conditions tested during the same experiment. The peak frequency of each population burst was measured as an indication of the potency of synchronic events. To analyse the responses to dorsal root stimulation, the firing of each individual neuron was measured during a 1 s time window before and after electrical stimulation of the dorsal root. Since most neurons showed spontaneous discharges, the data analysed were the difference in firing between both measurements. The post-stimulus response of some neurons during the ≈50 ms after the stimulus may not be accurately measured due to electrical artefacts in the signal and the accumulation of multi-unit firing in this time window (Roza et al., 2016). Due to the highly irregular and random firing of some neurons, the recordings were visually inspected to confirm that the effects were consistent. A selection was then made using numerical criteria. A neuron was considered to be excited when firing increased to 150% or more from control values. A decrease to less than 50% of control was considered inhibition.

In optogenetic studies, spontaneous activity was analysed in the absence of stimulation during 15 min for control responses and then the activity was analysed during the light pulses applied. For the ramps, the 30 s of activity after reaching the maximum intensity was analysed. For the hM4Di-DREADD studies, activity during 15 min of control and 10 min in CNO application (generally between 15 and 25 min after application onset) were measured, in both cases before dorsal root stimuli.

Statistical analyses and graphs were made using GraphPad Prism 7 (GraphPad Software Inc., La Jolla, CA, USA). Data are presented as mean ± SD. Comparisons between pairs of data were analysed by a Wilcoxon matched-pairs signed rank test. Comparisons between multiple values were made using one-way ANOVA followed by Dunnett's post-test. Correlograms were built to illustrate the relationship between sDRPs and the firing of action potentials by dorsal horn neurons. For these correlograms, the distribution of spikes recorded from dorsal horn neurons using MEAs was referred to the time of the peak of sDRP that was used as a reference (point 0 in graphs). When comparing two sets of data in the same correlogram, dorsal horn neuron firing was normalised dividing firing of dorsal horn neurons by the number of sDRPs analysed.

## Results

### Animal models and recording characteristics

To confirm the successful expression of rhodopsins and hM4Di-DREADD in primary afferents, mice were characterised by histological and behavioural methods and the results are summarised in Fig. 1. Cre-positive mice, as analysed by PCR, successfully expressed the reporter fluorescent proteins in the somata of primary afferent neurons at the dorsal root ganglia and their central terminals within the spinal cord. Further, light application to the dorsal surface of hind paws of channelrhodopsin-2-expressing mice evoked withdrawal reflexes demonstrating functional expression of the opsin.

Dorsal root recordings showed a stable basal potential and sDRPs at irregular intervals. Grand mean values for sDRP frequency and amplitude were 0.299 ± 0.159 Hz and 14.99 ± 6.35 µV respectively (data from 38 roots in 24 mice). Simultaneously recorded dorsal horn neurons showed spontaneous activity in the form of asynchronous low-frequency firing (ongoing component) and synchronous events or population bursts made up of action potentials fired by subsets of neurons. The grand mean frequency and peak amplitude for population bursts were 0.241 ± 0.135 Hz and 79.63 ± 36.49 Hz respectively (from 38 recording tracks in 24 mice). Population bursts

and sDRPs coincided in time, although not all sDRPs were associated with population bursts.

## Effects of light-induced depolarisation of primary afferents on the activity of dorsal roots and dorsal horn neurons

To test the effects of depolarisation of primary afferents, we used preparations from nine mice expressing channelrhodopsin-2. Light application produced direct depolarisation of primary afferents that was recorded from isolated nerves and dorsal roots. Isolated peripheral nerves (five nerves from two mice) stimulated with light pulses of constant intensity (455 nm, square pulses) produced A- and C-fibre volleys similar to those elicited by electrical stimulation (Fig. 2*A* and *B*).

Dorsal root recordings from spinal cord slices (Fig. 2*C*) showed fast and large-amplitude depolarisation peaks at the onset of high-intensity long-lasting square light pulses followed by a sustained depolarisation ($n = 7$ mice,

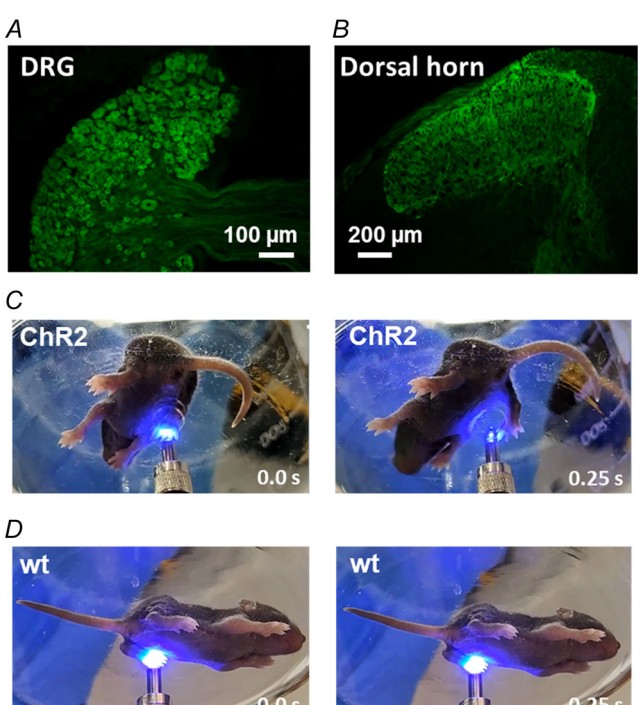

**Figure 1. Histological and behavioural characterisation of animal models**

*A* and *B*, photomicrographs showing EGFP expression in dorsal root ganglion neurons of all sizes (*A*) and their endings within the spinal cord (*B*) in an adult Arch/EGFP mouse. Note that EGFP labelling in the spinal cord covered both the dorsal horn and the medial intermediate area of the cord. *C*, light stimulation (455 nm) of the hind paws of neonatal mice expressing channelrhodopsin-2 (ChR2) produced withdrawal reflex-like responses. Light stimulation started at time 0 s (left frame). *D*, such behavioural responses were never observed in wild-type animals. [Colour figure can be viewed at wileyonlinelibrary.com]

Fig. 2*D*). Short-lasting pulses ($\leq 6$ ms) caused depolarising peaks (Fig. 2*E*) followed by a slow depolarisation exceeding the duration of the stimulus, probably caused by synaptic activation of spinal circuits (Fig. 2*E*). Application of light stimuli of increasing intensity (ramp stimuli) did not elicit fast peaks, but depolarisations of the dorsal root were clear and sustained (Fig. 2*F*).

The combined firing frequency of all neurons recorded from an electrode position increased clearly during light stimulation using short- and long-lasting pulses as well as ramp stimuli (Fig. 2*D*–*F*) indicating that volleys in primary afferents, but also a maintained level of depolarisation in the afferents, activated spinal cord neurons.

The effects of 60 s light pulses on spontaneous activity of dorsal roots and dorsal horn neurons are shown in Fig. 3 and quantified in Fig. 4. The amplitude of the light-induced depolarisation of the dorsal root and the frequency of the ongoing firing of dorsal horn neurons were directly related to light intensity and were maintained during the whole stimulation period (Figs 3*B* and 4*A*, *B*). However, light stimulation produced a strong inhibition of spontaneous dorsal root potentials and population bursts that virtually disappeared (Figs 3*C*, *E* and 4*C*, *D*). During light stimulation most neurons showed an increased spontaneous firing (409/773) but some were depressed (41/773) and others were not affected (323/773).

In six control experiments using preparations from Cre-negative littermates and wild-type mice, the application of light at 455 nm at maximum intensity did not induce any observable effect (Fig. 3*H*).

Responses to electrical stimulation of a dorsal root (eDRPs) were depressed by light-induced depolarisation of primary afferents as shown in Fig. 3*D* and *F* and quantified in Fig. 4*E* and *F*. The eDRPs were reduced by light stimulation in an intensity-dependent fashion. The response of dorsal horn neurons to electrical stimulation of the dorsal root was reduced during depolarisation of primary afferents in a context of increased ongoing activity.

## Effects of chemogenetic inhibition of primary afferents on the activity of dorsal root and dorsal horn neurons

The above results support the idea that central terminals of primary afferents may act as a synchronising element of spontaneous activity in spinal circuits. However, these results show that depolarisation of afferents produces some excitatory effects on ongoing firing of dorsal horn neurons probably due to spontaneous neurotransmitter release from the afferents. To test whether blockade of neurotransmitter release from primary afferents could prevent synchronous activity, further studies were carried

out in mice expressing hM4Di-DREADD under control of the advillin promoter. The hM4Di receptor was activated by bath application of 10 μM CNO. The results are summarised in Fig. 5.

Application of CNO did not produce any consistent alteration in the basal potential recorded from dorsal roots, but it had a strong inhibitory effect on the frequency of sDRPs (from $0.369 \pm 0.241$ to $0.042 \pm 0.053$ Hz after CNO; $P = 0.0313$, Wilcoxon matched-pairs signed rank test; $n = 6$). The sDRPs that remained had a reduced mean amplitude (from $20.7 \pm 9.3$ to $11.2 \pm 8.5$ μV after CNO; $P = 0.0313$, Wilcoxon matched-pairs signed rank test; $n = 6$). The concomitant population bursts recorded from dorsal horn neurons were also strongly reduced in frequency (from $0.342 \pm 0.215$ to $0.167 \pm 0.149$ Hz after CNO; $P = 0.0313$, Wilcoxon matched-pairs signed rank test; $n = 6$; see Fig. 5*B*) and amplitude (from $73.1 \pm 53.2$ to $42.0 \pm 47.9$ Hz after CNO; $P = 0.0313$, Wilcoxon matched-pairs signed rank test; $n = 6$). There was also a strong inhibition (<50% of control frequency) of spontaneous firing frequency in most of the recorded neurons (117 out of 186, 62.9%). Only 10 neurons showed an increase in firing frequency in the presence of CNO.

Dorsal root responses to electrical stimulation of a dorsal root were partially depressed in the presence of CNO (from $100 \pm 78$ to $53 \pm 55$ μV s; $P = 0.0313$, Wilcoxon matched-pairs signed rank test; $n = 6$; see Fig. 5*C*). Similarly, responses of dorsal horn neurons to electrical stimulation of the dorsal root were reduced in CNO (from $2.37 \pm 4.57$ spikes in control to $1.78 \pm 3.30$ spikes in the presence of CNO; $P < 0.001$, Wilcoxon matched-pairs signed rank test; $n = 186$).

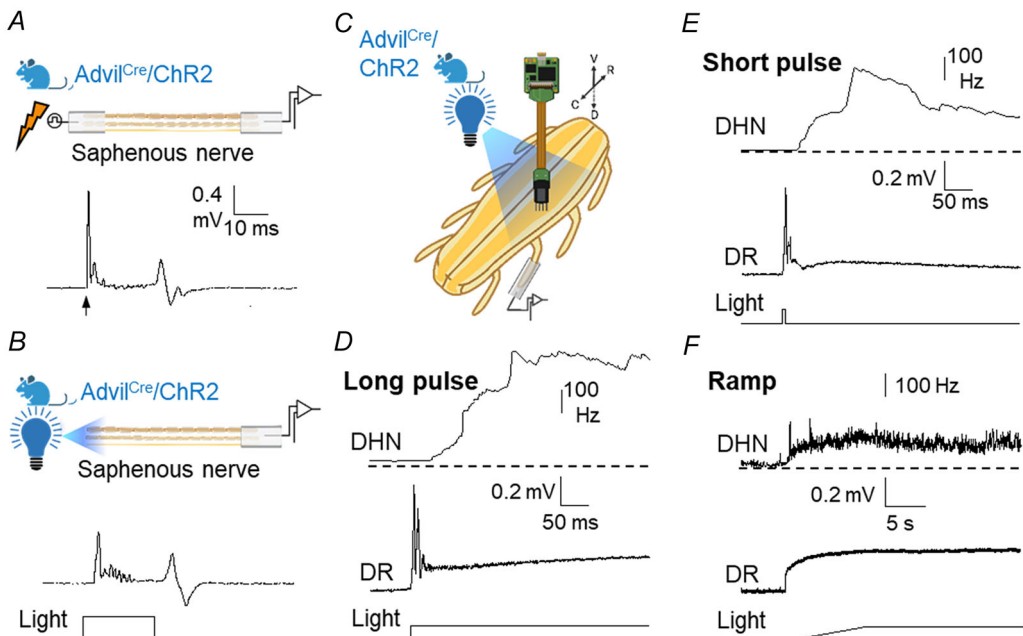

**Figure 2. Effects of light stimulation on primary afferents expressing ChR2**
*A* and *B*, the experimental arrangements to stimulate and record from a saphenous nerve (for this and the remaining figures, the bulb represents optical stimulation, and the electric ray symbol, electrical stimuli; open symbols are used for off and coloured for on). The trace in *A* shows the response recorded from the proximal end of a saphenous nerve (10 mm long; 22-day-old mouse) to a high-intensity electrical stimulus applied to the distal end which elicited A- and C-fibre volleys (the stimulus artefact was eliminated for clarity; marked by arrow). A similar response was elicited by light stimuli (*B*; 455 nm; intensity 42 mW/cm$^2$; calibration bars as in *A*). *C*, the experimental arrangement enabling optical stimulation of ChR2 expressed in primary afferents and simultaneous recording from a set of dorsal horn neurons and from a dorsal root using a longitudinal slice of the spinal cord. *D–F*, images showing responses to different light stimuli. In all cases, the combined firing from 70 dorsal horn neurons (labelled DHN; upper traces), the responses from the dorsal root (labelled DR; middle trace) and the characteristics of light stimuli applied (maximum intensity 42 mW/cm$^2$; lower traces) are shown. *D*, responses to a prolonged light pulse; *E*, responses to a brief light pulse. Square light pulses elicited fast peaks of activity in the dorsal root and firing in dorsal horn neurons that were followed by sustained lower-level activity. *F*, ramp stimuli did not evoke fast peaks but induced sustained firing of neurons and sustained depolarisation of primary afferents. [Colour figure can be viewed at wileyonlinelibrary.com]

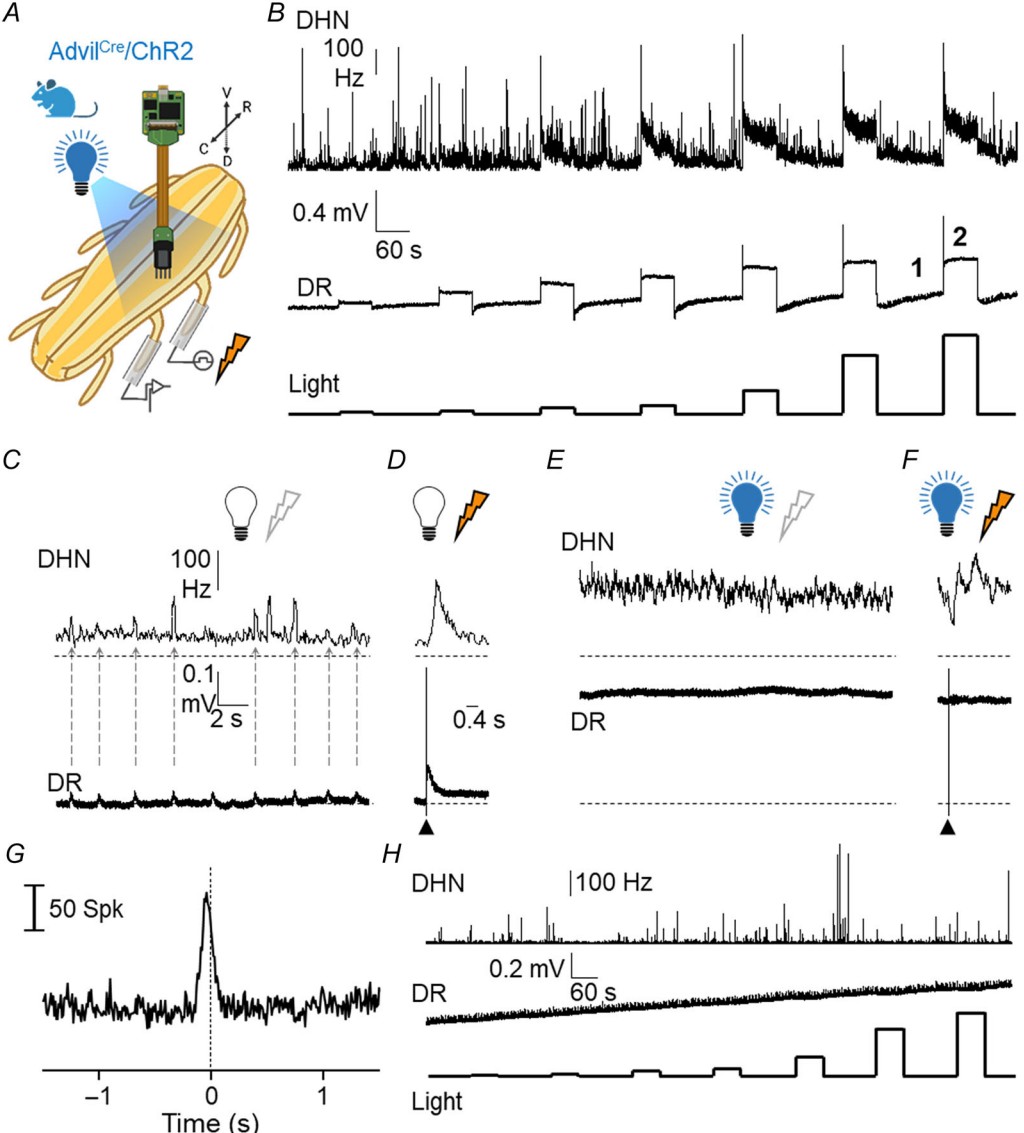

**Figure 3. Effects of ChR2 activation on dorsal roots and dorsal horn neurons**
*A*, the experimental set-up to allow for light activation of ChR2-expressing afferents and electrical stimulation of a dorsal root via a suction electrode while recording from an adjacent dorsal root and a set of dorsal horn neurons. *B*, the combined firing frequency of 70 dorsal horn neurons (DHN) and simultaneous dorsal root recordings (DR) during a prolonged period (22 min) with 60 s pulses of light-activation of ChR2 expressed in primary afferents (1–42 mW/cm²). Note the clear light intensity-dependent increases of firing frequency and sustained depolarisations. *C*, expanded recordings of the spontaneous activity in the absence of light (at point marked by 1 in *B*). Dotted arrows in *C* show the coincidence between spontaneous depolarisation of the dorsal root (DR) and spontaneous population bursts in dorsal horn neurons (DHN). *D*, the response to electrical stimulation (filled triangle) of the dorsal root in the absence of light. *E*, expanded recordings of the spontaneous activity in the presence of light (42 mW/cm²) at the point marked by 2 in *B*. Horizontal dotted lines in *E* correspond to the 0 level and traces show depolarised dorsal root recordings (DR) and high-frequency firing in dorsal horn neurons (DHN). Note the absence of sDRPs in the dorsal root and of population bursts in the combined activity of DHN. *F*, the absence of response to a high-intensity electrical stimulus delivered to the dorsal root (filled triangle) in the presence of light (42 mW/cm²). *G*, graph illustrating the distribution of action potentials from 70 neurons around the peak time of 328 sDRPs (time 0 and dotted line). The correlogram was built with data from a 15 min period in the absence of light. *H*, the combined firing frequency of 17 dorsal horn neurons (DHN) and simultaneous dorsal root recordings (DR) during a prolonged period (22 min) with 60 s pulses of light stimulation at 455 nm (1–42 mW/cm²) in a wild-type mouse. [Colour figure can be viewed at wileyonlinelibrary.com]

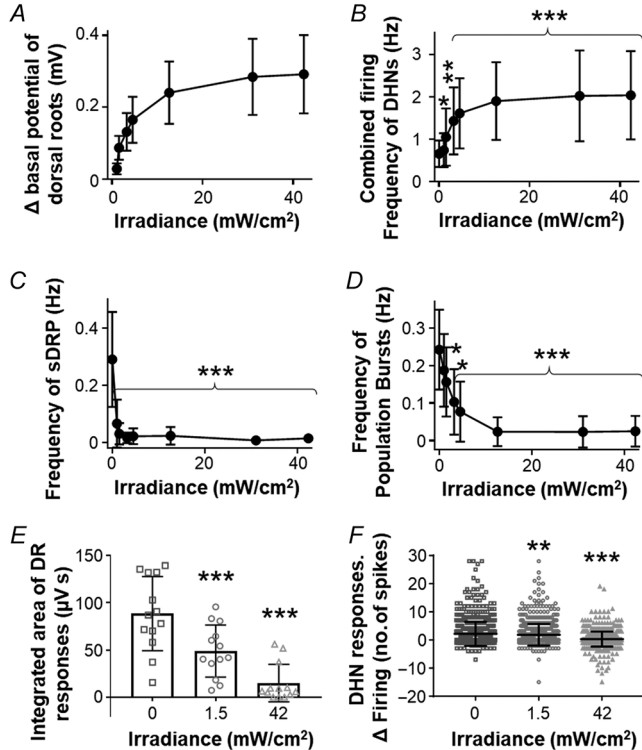

**Figure 4. Quantitative analysis of the effects of ChR2 activation on dorsal root and dorsal horn neurons**

*A–D*, intensity-dependent effects of light on depolarisation of dorsal roots (*A*), combined firing frequency of dorsal horn neurons [*B*; one-way ANOVA followed by Dunnett's post-test; $F_{(1.352, 16.23)} = 31.12$; 0 *vs.* 1 mW/cm$^2$ $P = 0.351$, 0 *vs.* 1.5 mW/cm$^2$ $P = 0.0442$, 0 *vs.* 3.2 mW/cm$^2$ $P = 0.0028$, and $P < 0.001$ from 4.5 mW/cm$^2$ when compared to baseline], frequency of spontaneous DRPs in dorsal roots [*C*; one-way ANOVA followed by Dunnett's post-test; $F_{(1.493, 17.91)} = 28.04$, $P < 0.001$ for all light intensities comparing with no light stimuli] and frequency of population bursts recorded from dorsal horn neurons [*D*; one-way ANOVA followed by Dunnett's post-test; $F_{(2.639, 31.67)} = 29.05$; 0 *vs.* 1 mW/cm$^2$ $P = 0.0108$, 0 *vs.* 1.5 mW/cm$^2$ $P = 0.0317$, and $P < 0.001$ from 3.2 mW/cm$^2$ when compared with no light stimuli] (*n* = 13, obtained from seven mice). *E* and *F*, graphs illustrating the inhibitory effect of ChR2 activation on responses elicited by electrical stimulation of a dorsal root. The area of the eDRPs was strongly reduced during light-induced depolarisation [*E*; one-way ANOVA followed by Dunnett's post-test; $F_{(1.176, 14.11)} = 47.02$; $P < 0.001$; *n* = 13, from seven mice]. *F*, the changes in evoked firing of each individual neuron during light stimulation. A 1 s time window before and after the electrical stimulation of the dorsal root was measured. Data shown are the difference in firing between both measurements. The firing of dorsal horn neurons was significantly depressed [one-way ANOVA followed by Dunnett's post-test; $F_{(1.654, 1277)} = 112.1$; 0 *vs.* 1.5 mW/cm$^2$ $P = 0.0021$, 0 *vs.* 42 mW/cm$^2$ $P < 0.001$; *n* = 773 neurons recorded from 13 tracks in seven mice]. Asterisks indicate statistically significant differences (*$P < 0.05$, **$P < 0.01$ and ***$P < 0.001$).

Using control mice (Cre-negative littermates, $n = 2$), application of CNO at the same concentration had no effect on dorsal root recordings and dorsal horn neurons.

## Effects of light-induced hyperpolarisation of primary afferents on the activity of dorsal roots and dorsal horn neurons

Since depolarisation of primary afferents with channelrhodopsin-2 and their inhibition by hM4Di-DREADD prevented synchronisation of spontaneous activity in dorsal horn neurons, we hypothesised that hyperpolarisation of the afferents should have the opposite effect. To test this idea, mice expressing the light-activated proton pump archaerhodopsin-3 under control of the advillin promoter were used. The main results obtained are summarised in Fig. 6 and analysed in Fig. 7.

Application of square light pulses at 567 nm produced an intensity-dependent hyperpolarisation of the basal potential recorded from the dorsal root in all preparations tested (12 dorsal roots, six mice). The response to light began with a rapid fall of the basal potential, which showed some degree of adaptation but remained hyperpolarised for the entire duration of the light pulse (Fig. 6*B*).

Both sDRPs and concomitant population bursts were potentiated by light in an intensity-dependent manner (Figs 6 and 7). Both events showed large increases in amplitude and frequency during hyperpolarisation (see Fig. 7). Spontaneous activity of dorsal horn neurons increased mostly due to the increase in amplitude and frequency of population bursts (see Fig. 7*D* and *F*). About half of neurons recorded (217/410) showed a slight increase in spontaneous activity.

Dorsal root and neuronal responses to electrical stimulation of an adjacent root were examined in the absence and presence of light pulses (Fig. 6). The integrated area of eDRPs increased during light-induced hyperpolarisation (Fig. 7*G*). In contrast, the responses of dorsal horn neurons to electrical stimuli were not significantly altered (Fig. 7*H*).

In control experiments with Cre-negative animals there were no visible responses to light stimulation in dorsal roots or dorsal horn neurons (five dorsal roots, three mice, 148 neurons; see Fig. 6*I*).

## Discussion

The results presented in this study show that modifying the membrane potential of the central terminals of primary afferents has a significant impact on the spontaneous activity exhibited by spinal cord circuits *in vitro*. We have previously shown that the spontaneous activity of primary afferents and dorsal horn neurons

is highly correlated, with firing of dorsal horn neurons occurring both before and after the activity in primary afferents (Lucas-Romero et al., 2022). The present work demonstrates that afferents do play a fundamental role in synchronising basal activity within dorsal horn circuits, helping to maintain a level of excitability in the absence of external inputs. The present results were obtained by manipulating the entire population of primary afferents; future studies targeting subpopulations of primary afferents may allow the contribution of specific afferents to the coordination of spontaneous activity in the dorsal horn to be defined.

At the spinal cord level, primary afferent depolarisation constitutes a fundamental mechanism for controlling incoming information. Primary afferent depolarisation is initiated by the activation of presynaptic receptors, with GABA-A receptors being the primary but not exclusive system (Zimmerman et al., 2019). While the conventional view defends that depolarisation serves for presynaptic inhibition (Rudomin & Schmidt, 1999), a recent study on proprioceptive afferents described the facilitation of spike propagation at branching points by GABA-A receptor activation (Hari et al., 2022). In other brain areas, GABA-A receptors located at the axonal level constitute a potent regulatory mechanism, capable of producing either excitatory or inhibitory actions on neurotransmitter release (Trigo et al., 2008).

In our experiments, light-induced depolarisation produced a complete abolition of sDRPs in dorsal roots. Considering that sDRPs are tightly associated with population bursts, abolition of population bursts could be interpreted as an effect secondary to the abolition of sDRPs. Furthermore, during light-induced depolarisation, the responses to electrical stimulation of the dorsal roots recorded from dorsal horn neurons and from adjacent dorsal roots were depressed. However, these depressant effects occurred despite a net increase in the ongoing firing of many dorsal horn neurons, probably due to an enhanced spontaneous release of neurotransmitters from primary afferents.

Excitatory actions on neurotransmitter release due to the depolarisation of axons has been previously described in other brain areas (Jang et al., 2001, 2002; Ruiz et al., 2010; Zorrilla de San Martin et al., 2017). Depolarisation-induced release of neurotransmitters may be due to an increase in calcium levels in synaptic terminals (Awatramani et al., 2005). A depression of neurotransmitter release induced by action potentials in a context of increased spontaneous release has been previously reported in spinal cord (Jang et al., 2002) and pituitary neurons (Zhang & Jackson, 1993). Considering all these aspects, a plausible explanation for the observed results is that light-induced depolarisation interferes with action potential-dependent synaptic transmission but favours the spontaneous release of neurotransmitters, thereby increasing the ongoing firing of dorsal horn neurons. The depression of population bursts, a form of coordinated spontaneous activity in dorsal horn neurons, may therefore indicate a dependency on action potential firing in primary afferent terminals. Primary afferent depolarisation produced by GABA-A receptor activation has traditionally been associated with pre-synaptic inhibition. Several mechanisms have been proposed to mediate the inhibitory effects of GABA. Depolarisation-induced inactivation of voltage-gated sodium and/or calcium channels may reduce calcium

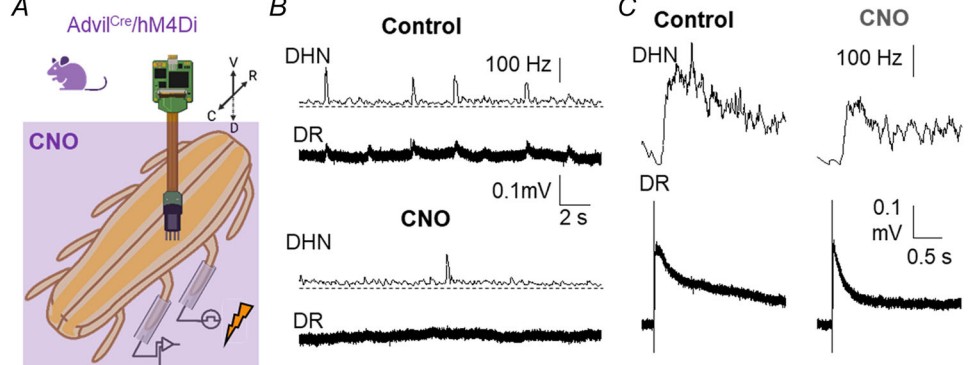

**Figure 5. Effects of hM4Di-DREADD activation on dorsal roots and dorsal horn neurons**
*A*, the arrangement used for the experiments. Simultaneous recordings were obtained from the primary afferents contained in the dorsal root (DR) using suction electrodes and from dorsal horn neurons (DHN) by means of multi-electrode arrays. Electrical stimuli were applied to an adjacent dorsal root using suction electrodes, and CNO at 10 μM was bath applied to the entire preparation. *B*, an original recording of spontaneous activity from a representative experiment under control and CNO conditions. In comparison with control, application of CNO depressed sDRPs (DR) and population bursts of dorsal horn neurons (DHN). *C*, electrical stimulation of an adjacent dorsal root elicited a DRP that was also reduced by the activation of hM4Di-DREADD. In addition, the evoked firing combined from 26 dorsal horn neurons was reduced in the presence of CNO. [Colour figure can be viewed at wileyonlinelibrary.com]

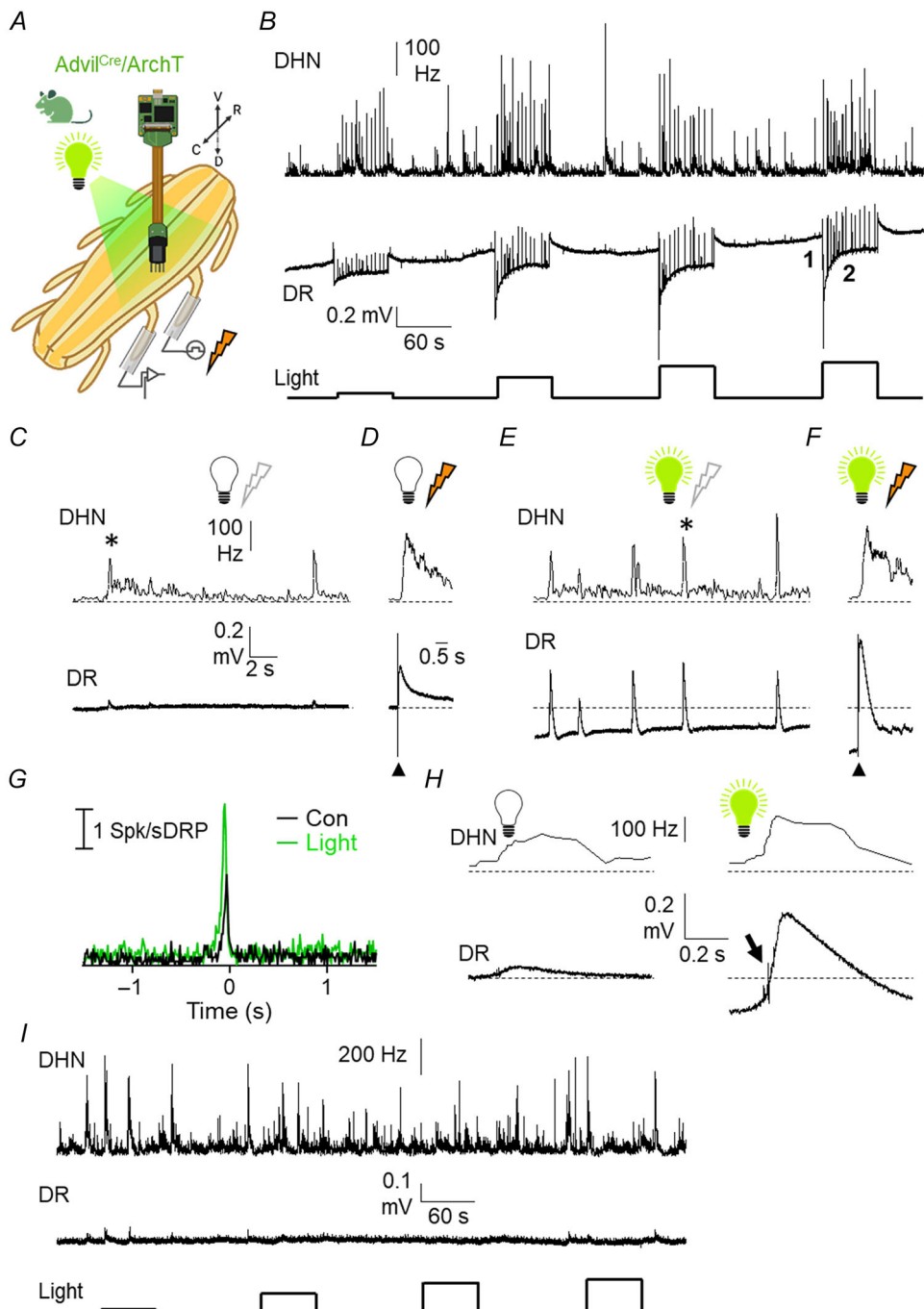

**Figure 6. Effects of light-induced activation of archaerhodopsin-3 (ArchT) on dorsal roots and dorsal horn neurons**

*A*, the experimental arrangement, identical to that of Fig. 3 except for donor mice (here ArchT-expressing) and light wavelength (here 567 nm). *B*, the combined firing frequency of 41 dorsal horn neurons (DHN) and simultaneous dorsal root recordings (DR) during a prolonged period (12 min) with 60 s pulses of light stimuli of different intensities (from 30 to 187 mW/cm²). Note the clear light intensity-dependent hyperpolarisation of the dorsal root, the increase in amplitude of sDRPs and the concomitant gain in amplitude of population bursts in the combined firing of dorsal neurons. *C*, expanded recording of the spontaneous activity in the absence of light (marked by 1 in *B*). *D*, responses obtained during electrical dorsal stimulation in the absence of light. The filled triangle marks the delivery of a high-intensity electrical stimulus to the dorsal root and responses recorded from neurons and the adjacent dorsal root. *E*, as for *C* in the presence of light (187 mW/cm²) (trace obtained at time marked by 2 in *B*). Horizontal dotted lines in *E* correspond to the 0 level and traces show hyperpolarised dorsal root recordings (DR) with high-amplitude sDRPs and concomitant large population bursts in dorsal horn neurons (DHN). *F*, increased

dorsal root response to a high-intensity electrical pulse delivered to the dorsal root (filled triangle) in the presence of light (187 mW/cm$^2$) and the lack of change in the combined firing of DHN. *G*, normalized correlograms between sDRPs and spikes in the set of dorsal horn neurons. Graph shows the distribution of action potentials from 41 neurons around the time peaks of sDRPs (time 0) in the absence (control trace; 10 sDRPs) and presence (light trace; 18 sDRPs) of light (187 mW/cm$^2$). *H*, a comparison between spontaneous events in the absence and presence of light (187 mW/cm$^2$). During the light-induced hyperpolarisation, large spikes were detected at the rising phase of the sDRP (arrow) that may correspond with the firing of primary afferents. *I*, a control experiment for light stimulation at 567 nm (from 30 to 187 mW/cm$^2$) in a Cre-negative mouse. Image shows the combined firing frequency of 44 dorsal horn neurons (DHN) and simultaneous dorsal root recordings (DR) during 60 s pulses. [Colour figure can be viewed at wileyonlinelibrary.com]

entry into the presynaptic terminal and synaptic transmission (French et al., 2006; Rudomin & Schmidt, 1999). The shunting effect produced by GABA-A receptor conductance has been also proposed to reduce action potential size and synaptic release (Cattaert & El Manira, 1999). The implication of these mechanisms has been tested using biological and computational models with mixed results (Cattaert & El Manira, 1999; French et al., 2006; Graham & Redman, 1994; Walmsley et al., 1995). Other forms of inhibition involving presynaptic terminals include post-activation depression. This form of inhibition can last for several seconds and may occur because of a reduction in available neurotransmitters, reduced excitability of primary afferents and/or activation of synaptic inhibitory mechanisms such as GABA-B receptors (Metz et al., 2023).

To ascertain whether a reduction in the excitability of primary afferents hinders coordination in spinal circuits, an alternative approach using an inhibitory DREADD was implemented. Activation of DREADD with CNO strongly reduced sDRPs and population bursts in all preparations examined. Additionally, CNO application resulted in a marked decrease in spontaneous firing in dorsal horn neurons, indicating that afferent activity is crucial for maintaining a basal level of activity in spinal circuits. These results provide further support for the role of primary afferents in the expression of coordinated spontaneous activity. The hM4Di-DREADD stimulates inhibitory G-proteins inhibiting the adenylate cyclase pathway, potentially affecting several downstream targets. Blockade of neurotransmitter release has been proposed as a mechanism of action for hM4Di-DREADD (Stachniak et al., 2014). The reduction in responses to electrical stimulation is consistent with an inhibition of transmitter release, but complete inhibition of transmission was not observed under our experimental conditions. When expressed in TRPV1 afferents, hM4Di-DREADD activation has been shown to elevate heat threshold *in vivo* by activating potassium currents (Saloman et al., 2016). In our experiments the basal potential of afferents was not affected by CNO, indicating that potassium channels are not involved.

The hyperpolarisation of primary afferents by light-activation of archaerhodopsin-3 increased the amplitude and frequency of sDRPs and enhanced the coordination of spontaneous activity of dorsal horn neurons giving rise to larger population bursts. This observation further confirms the causal relationship between sDRPs and population bursts. During hyperpolarisation pulses, spontaneous firing of dorsal horn neurons was augmented because of the increase in frequency and magnitude of population bursts. This effect differs from the increase in ongoing firing in the absence of population bursts observed during depolarisation of the afferents with ChR2. Light-induced hyperpolarisation of primary afferents produced in our experiments may be large considering previous reports (Daou et al., 2016). A large hyperpolarisation should enhance the drive for depolarisation in response to synaptic activation of GABA-A receptors. As a consequence, both eDRPs and sDRPs were augmented. An intense depolarisation may also facilitate the firing of primary afferents during the rising phase of the sDRP, thereby amplifying the firing associated with population bursts. In previous studies, we reported an increase of spontaneous activity in dorsal roots and population bursts after inflammation of the hind paw (Lucas-Romero et al., 2022; Vicente-Baz et al., 2022). Interestingly, other studies indicate that, in a first instance, intense nociceptive stimuli produce hyperpolarisation of slowly conducting afferents, facilitating nociceptive transmission (Ramirez-Morales et al., 2021), and this effect may be involved in the changes observed during inflammation. In other brain areas, axonal hyperpolarisation is thought to facilitate transmitter release due to sodium channel recovery from inactivation and increased calcium influx (Rama et al., 2015; Zbili & Debanne, 2019). In the spinal cord, the primary afferent hyperpolarisation that follows a tetanic stimulus *in vivo* increases synaptic efficacy (Lloyd, 1949; Mendell & Wall, 1964). However, during light-induced hyperpolarisation, the responses of dorsal horn neurons to electrical stimulation of the dorsal root were not enhanced as would be expected. Primary afferents are likely to be more depolarised under *in vivo* conditions than *in vitro* (Mendell & Wall, 1964). It is possible that the mechanism involved in the hyperpolarisation-induced increase in neurotransmitter release requires a tonically depolarised potential, which is not present in isolated conditions such as *in vitro* preparations. In addition, the recruitment of primary afferents in response to electrical stimulation

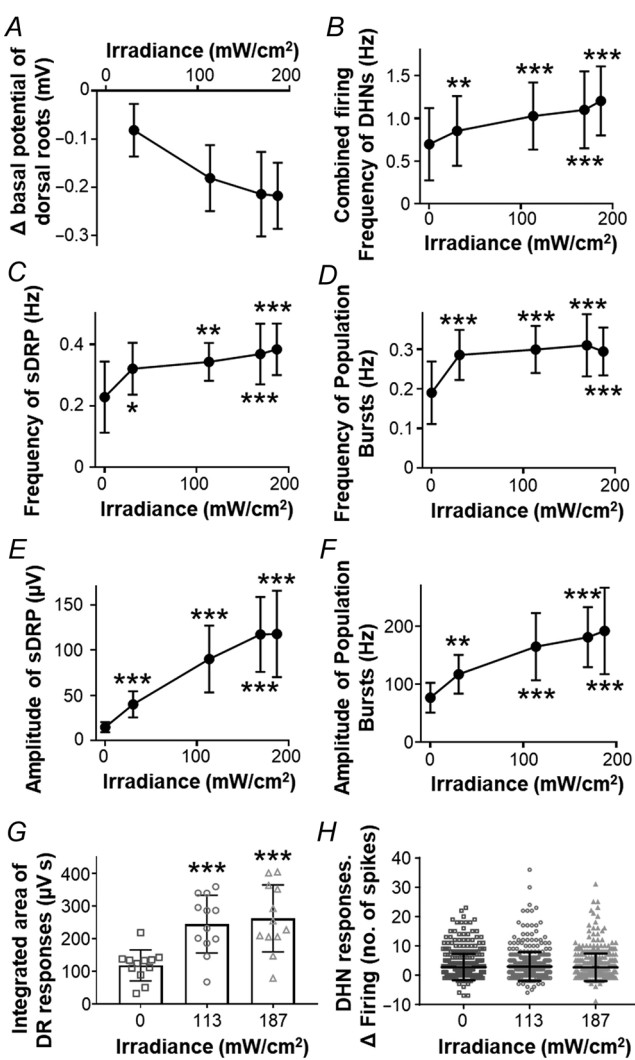

eDRP wave increased during light stimulation [*G*; one-way ANOVA followed by Dunnett's post-test; $F_{(1.517, 16.68)} = 47.43$; $P < 0.001$; $n = 12$, obtained from six mice]. *H*, the changes in evoked firing of each individual neuron during light stimulation. A 1 s time window before and after the electrical stimulation of the dorsal root was measured. Data shown are the difference in firing between both measurements. The firing of dorsal horn neurons ($n = 410$ neurons recorded from 12 tracks in six mice) was unaffected by light-induced hyperpolarisation of the afferents [one-way ANOVA followed by Dunnett's post-test; $F_{(1.958, 800.7)} = 0.1047$; $P = 0.9987$ and $P = 0.8833$ for 113 and 187 mW/cm$^2$ respectively]. Asterisks indicate statistically significant differences (*$P < 0.05$, **$P < 0.01$ and ***$P < 0.001$).

may be saturated under basal conditions and hence no further enhancement in the responses should be expected during hyperpolarisation. A final factor to consider is the effect of archaerhodopsin on synaptic transmission. The proton pumping activity of archaerhodopsin can led to alkalinisation of synaptic terminals, resulting in an increase in spontaneous neurotransmitter release and silencing of evoked transmission (El-Gaby et al., 2016; Mahn et al., 2016). Although alkalinisation develops slowly during archaerhodopsin activation and this does not correlate with the fast actions of light on primary afferent responses, it may contribute to some extent to the observed effects.

Collectively, the findings presented here underscore the essential role of primary afferents in coordinating spontaneous activity of dorsal horn neurons to produce population bursts. Firing of the afferents appears to contribute decisively to the population bursts recorded from dorsal horn neurons. These events are likely to play a fundamental role in maintaining the excitability of spinal cord circuits at adequate levels for optimal readiness conditions and to optimise the reception and integration of sensory signals from the periphery. Furthermore, this study directly highlights the critical function of the central terminals of primary afferents as a control mechanism for regulating the entry of somatosensory information into the CNS.

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

**Figure 7. Quantitative analysis of effects of ArchT activation on dorsal roots and dorsal horn neurons**
*A–F*, quantitative effects produced by light activation of ArchT at different irradiances on the parameters analysed for spontaneous activity ($n = 12$, obtained from six mice). The basal potential of the dorsal root was hyperpolarised by light stimulation (*A*). The spontaneous firing frequency of dorsal horn neurons increased [*B*; one-way ANOVA followed by Dunnett's post-test; $F_{(2.764, 30.4)} = 29.46$; 0 *vs.* 30.5 mW/cm$^2$ $P = 0.0033$, $P < 0.001$ from 113.5 mW/cm$^2$ when compared with no light stimuli]. The frequency of sDRPs was higher in the presence of light [*C*; one-way ANOVA followed by Dunnett's post-test; $F_{(2.422, 26.64)} = 16.61$; 0 *vs.* 30.5 mW/cm$^2$ $P = 0.0125$, $P < 0.001$ from 113.5 mW/cm$^2$ compared with no light]. The frequency of population bursts also increased [*D*; one-way ANOVA followed by Dunnett's post-test; $F_{(3.055, 33.61)} = 22.3$; $P < 0.001$ for all comparisons with zero light]. The amplitude of sDRPs augmented during light stimulation [*E*; one-way ANOVA followed by Dunnett's post-test; $F_{(1.438, 15.82)} = 56.74$; $P < 0.001$]. Finally, the amplitude of population bursts was higher with light [*F*; one-way ANOVA followed by Dunnett's post-test; $F_{(2.109, 23.2)} = 30.39$; 0 *vs.* 30.5 mW/cm$^2$ $P = 0.0012$, $P < 0.001$ from 113.5 mW/cm$^2$ compared with no light]. *G* and *H*, bar graphs illustrating the effect of light activation of ArchT at 113 and 187 mW/cm$^2$ on the responses elicited by the electrical stimulation of an adjacent dorsal root. The area of the

Cattaert, D., & El Manira, A. (1999). Shunting versus inactivation: Analysis of presynaptic inhibitory mechanisms in primary afferents of the crayfish. *Journal of Neuroscience*, **19**(14), 6079–6089.

Daou, I., Beaudry, H., Ase, A. R., Wieskopf, J. S., Ribeiro-da-Silva, A., Mogil, J. S., & Seguela, P. (2016). Optogenetic silencing of Nav1.8-positive afferents alleviates inflammatory and neuropathic pain. *eNeuro*, **3**(1), https://doi.org/10.1523/ENEURO.0140-15.2016

Eccles, J. C., Eccles, R. M., & Magni, F. (1961). Central inhibitory action attributable to presynaptic depolarization produced by muscle afferent volleys. *The Journal of Physiology*, **159**(1), 147–166.

El-Gaby, M., Zhang, Y., Wolf, K., Schwiening, C. J., Paulsen, O., & Shipton, O. A. (2016). Archaerhodopsin selectively and reversibly silences synaptic transmission through altered pH. *Cell Reports*, **16**(8), 2259–2268.

French, A. S., Panek, I., & Torkkeli, P. H. (2006). Shunting versus inactivation: Simulation of GABAergic inhibition in spider mechanoreceptors suggests that either is sufficient. *Neuroscience Research*, **55**(2), 189–196.

Graham, B., & Redman, S. (1994). A simulation of action potentials in synaptic boutons during presynaptic inhibition. *Journal of Neurophysiology*, **71**(2), 538–549.

Hari, K., Lucas-Osma, A. M., Metz, K., Lin, S., Pardell, N., Roszko, D. A., Black, S., Minarik, A., Singla, R., Stephens, M. J., Pearce, R. A., Fouad, K., Jones, K. E., Gorassini, M. A., Fenrich, K. K., Li, Y., & Bennett, D. J. (2022). GABA facilitates spike propagation through branch points of sensory axons in the spinal cord. *Nature Neuroscience*, **25**(10), 1288–1299.

Inacio, A. R., Nasretdinov, A., Lebedeva, J., & Khazipov, R. (2016). Sensory feedback synchronizes motor and sensory neuronal networks in the neonatal rat spinal cord. *Nature Communications*, **7**(1), 13060.

Jang, I. S., Jeong, H. J., Katsurabayashi, S., & Akaike, N. (2002). Functional roles of presynaptic GABA(A) receptors on glycinergic nerve terminals in the rat spinal cord. *The Journal of Physiology*, **541**(2), 423–434.

Jang, I. S., Jeong, H. J., & Akaike, N. (2001). Contribution of the Na-K-Cl cotransporter on GABA(A) receptor-mediated presynaptic depolarization in excitatory nerve terminals. *Journal of Neuroscience*, **21**(16), 5962–5972.

Khatri, S. N., Wu, W., Yang, Y., & Pugh, J. R. (2019). Direction of action of presynaptic GABA(A) receptors is highly dependent on the level of receptor activation. *Journal of Neurophysiology*, **121**(5), 1896–1905.

Kramer, P. F., Twedell, E. L., Shin, J. H., Zhang, R., & Khaliq, Z. M. (2020). Axonal mechanisms mediating gamma-aminobutyric acid receptor type A (GABA-A) inhibition of striatal dopamine release. *eLife*, **9**, e55729.

Li, J., & Baccei, M. L. (2011). Pacemaker neurons within newborn spinal pain circuits. *Journal of Neuroscience*, **31**(24), 9010–9022.

Lloyd, D. P. C. (1949). Post-tetanic potentiation of response in monosynaptic reflex pathways of the spinal cord. *Journal of General Physiology*, **33**(2), 147–170.

Lucas-Romero, J., Rivera-Arconada, I., & Lopez-Garcia, J. A. (2024). Noise or signal? Spontaneous activity of dorsal horn neurons: Patterns and function in health and disease. *Pflugers Archiv: European Journal of Physiology*, **476**(8), 1171–1186.

Lucas-Romero, J., Rivera-Arconada, I., & Lopez-Garcia, J. A. (2022). Synchronous firing of dorsal horn neurons at the origin of dorsal root reflexes in naive and paw-inflamed mice. *Frontiers in Cellular Neuroscience*, **16**, 1004956.

Luz, L. L., Szucs, P., & Safronov, B. V. (2014). Peripherally driven low-threshold inhibitory inputs to lamina I local-circuit and projection neurones: A new circuit for gating pain responses. *The Journal of Physiology*, **592**(7), 1519–1534.

Mahn, M., Prigge, M., Ron, S., Levy, R., & Yizhar, O. (2016). Biophysical constraints of optogenetic inhibition at presynaptic terminals. *Nature Neuroscience*, **19**(4), 554–556.

Mendell, L. M., & Wall, P. D. (1964). Presynaptic hyperpolarization: A role for fine afferent fibres. *The Journal of Physiology*, **172**(2), 274–294.

Metz, K., Matos, I. C., Hari, K., Bseis, O., Afsharipour, B., Lin, S., Singla, R., Fenrich, K. K., Li, Y., Bennett, D. J., & Gorassini, M. A. (2023). Post-activation depression from primary afferent depolarization (PAD) produces extensor H-reflex suppression following flexor afferent conditioning. *The Journal of Physiology*, **601**(10), 1925–1956.

Pachitariu, M., Sridhar, S., Pennington, J., & Stringer, C. (2024). Spike sorting with Kilosort4. *Nature Methods*, **21**(5), 914–921.

Price, T. J., Cervero, F., Gold, M. S., Hammond, D. L., & Prescott, S. A. (2009). Chloride regulation in the pain pathway. *Brain Research Reviews*, **60**(1), 149–170.

Rama, S., Zbili, M., Bialowas, A., Fronzaroli-Molinieres, L., Ankri, N., Carlier, E., Marra, V., & Debanne, D. (2015). Presynaptic hyperpolarization induces a fast analogue modulation of spike-evoked transmission mediated by axonal sodium channels. *Nature Communications*, **6**(1), 10163.

Ramirez-Morales, A., Hernandez, E., & Rudomin, P. (2021). Nociception induces a differential presynaptic modulation of the synaptic efficacy of nociceptive and proprioceptive joint afferents. *Experimental Brain Research*, **239**(8), 2375–2397.

Rivera-Arconada, I., & Lopez-Garcia, J. A. (2006). Retigabine-induced population primary afferent hyperpolarisation *in vitro*. *Neuropharmacology*, **51**, 756–763.

Roza, C., Mazo, I., Rivera-Arconada, I., Cisneros, E., Alayon, I., & Lopez-Garcia, J. A. (2016). Analysis of spontaneous activity of superficial dorsal horn neurons *in vitro*: Neuropathy-induced changes. *Pflugers Archiv: European journal of physiology*, **468**(11–12), 2017–2030.

Rudomin, P., & Schmidt, R. F. (1999). Presynaptic inhibition in the vertebrate spinal cord revisited. *Experimental Brain Research*, **129**(1), 1–37.

Ruiz, A., Campanac, E., Scott, R. S., Rusakov, D. A., & Kullmann, D. M. (2010). Presynaptic GABAA receptors enhance transmission and LTP induction at hippocampal mossy fiber synapses. *Nature Neuroscience*, **13**(4), 431–438.

Saloman, J. L., Scheff, N. N., Snyder, L. M., Ross, S. E., Davis, B. M., & Gold, M. S. (2016). Gi-DREADD expression in peripheral nerves produces Ligand-dependent analgesia, as well as Ligand-independent functional changes in sensory neurons. *Journal of Neuroscience*, **36**(42), 10769–10781.

Stachniak, T. J., Ghosh, A., & Sternson, S. M. (2014). Chemogenetic synaptic silencing of neural circuits localizes a hypothalamus→midbrain pathway for feeding behavior. *Neuron*, **82**(4), 797–808.

Trigo, F. F., Marty, A., & Stell, B. M. (2008). Axonal GABAA receptors. *European Journal of Neuroscience*, **28**(5), 841–848.

Vicente-Baz, J., Lopez-Garcia, J. A., & Rivera-Arconada, I. (2022). Central sensitization of dorsal root potentials and dorsal root reflexes: An *in vitro* study in the mouse spinal cord. *European Journal of Pain (London, England)*, **26**(2), 356–369.

Walmsley, B., Graham, B., & Nicol, M. J. (1995). Serial E-M and simulation study of presynaptic inhibition along a group ia collateral in the spinal cord. *Journal of Neurophysiology*, **74**(2), 616–623.

Wang, L., Kloc, M., Maher, E., Erisir, A., & Maffei, A. (2019). Presynaptic GABAA receptors modulate thalamocortical inputs in layer 4 of rat V1. *Cerebral Cortex*, **29**(3), 921–936.

Zbili, M., & Debanne, D. (2019). Past and future of analog-digital modulation of synaptic transmission. *Frontiers in Cellular Neuroscience*, **13**, 160.

Zhang, S. J., & Jackson, M. B. (1993). GABA-activated chloride channels in secretory nerve endings. *Science*, **259**(5094), 531–534.

Zimmerman, A. L., Kovatsis, E. M., Pozsgai, R. Y., Tasnim, A., Zhang, Q., & Ginty, D. D. (2019). Distinct modes of presynaptic inhibition of cutaneous afferents and their functions in behavior. *Neuron*, **102**(2), 420–434.e8.

Zorrilla de San Martin, J., Trigo, F. F., & Kawaguchi, S. (2017). Axonal GABA(A) receptors depolarize presynaptic terminals and facilitate transmitter release in cerebellar Purkinje cells. *The Journal of Physiology*, **595**(24), 7477–7493.

# Additional information

### Data availability statement

All data are archived and fully available upon reasonable request. Data used for the analysis and construction of Figs 4 and 7 are available as supporting information. The data will be incorporated in the public repository of the Universidad de Alcala upon completion of the project.

### Competing interests

The authors declare that they have no conflicts of interest.

### Author contributions

J.L.-R. participated in the conception and design of the work, in the acquisition and analysis of the data, and revised the final version of the work. J.A.L.-G. contributed to the conception and design of the work, the interpretation of data, and drafting the work and revising it critically for important intellectual content. I.R.-A. participated in the conception and design of the work, in the acquisition, analysis and interpretation of data as well as drafting the work. All authors approved the final version of the manuscript; agree to be accountable for all aspects of the work in ensuring that questions related to the accuracy or integrity of any part of the work are appropriately investigated and resolved; and all persons designated as authors qualify for authorship, and all those who qualify for authorship are listed.

### Funding

The research leading to these results received funding from the Spanish Ministry of Science and Innovation (grant PID2021-126330OB-I00 funded by MCIN/AEI /https://doi.org/10.13039/501100011033 and by 'ERDF A way of making Europe', by the European Union).

### Acknowledgements

The authors would like to thank the Photonics Engineering Group of the University of Alcala for their help in measuring the light irradiance of the devices used in this work.

### Author's present address

J. Lucas-Romero: Department of Physical Therapy, Washington University School of Medicine, Washington University in St. Louis, St. Louis, MO 63108, USA.

### Keywords

dorsal horn neuron, optogenetics, primary afferent depolarisation, spontaneous activity, synchronous firing

# Supporting information

Additional supporting information can be found online in the Supporting Information section at the end of the HTML view of the article. Supporting information files available:

**Peer Review History**
**Supporting Information**

