## [Peer Review History · The Journal of Physiology]

Central terminals of primary afferents coordinate the spontaneous activity of dorsal horn neurons.

Javier Lucas-Romero, Jose Antonio Lopez-Garcia, and Ivan Rivera-Arconada
DOI: 10.1113/JP287970

Corresponding author(s): *Ivan Rivera-Arconada (ivan.rivera@uah.es)*

The following individual(s) involved in review of this submission have agreed to reveal their identity: *David J Bennett (Referee #2)*

Review Timeline:

Submission Date:	28-Oct-2024
Editorial Decision:	10-Dec-2024
Revision Received:	05-May-2025
Accepted:	21-May-2025

Senior Editor: *Nathan Schoppa*

Reviewing Editor: *Nathan Schoppa*

Transaction Report:

Dear Dr Rivera-Arconada,

Re: JP-RP-2024-287970 "Central terminals of primary afferents coordinate the spontaneous activity of dorsal horn neurons." by Javier Lucas-Romero, Jose Antonio Lopez-Garcia, and Ivan Rivera-Arconada

Thank you for submitting your manuscript to The Journal of Physiology. It has been assessed by a Reviewing Editor and by 2 expert referees and we are pleased to tell you that it is potentially acceptable for publication following satisfactory major revision.

REVISION CHECKLIST:

Please upload two versions of your manuscript text: one with all relevant changes highlighted and one clean version with no

changes tracked. The manuscript file should include all tables and figure legends, but each figure/graph should be uploaded as separate, high-resolution files.

We look forward to receiving your revised submission.

Yours sincerely,

Nathan Schoppa
Senior Editor
The Journal of Physiology

REQUIRED ITEMS

- Please include an Abstract Figure file, as well as the Figure Legend text within the main article file. The Abstract Figure is a piece of artwork designed to give readers an immediate understanding of the research and should summarise the main conclusions. If possible, the image should be easily 'readable' from left to right or top to bottom. It should show the physiological relevance of the manuscript so readers can assess the importance and content of its findings. Abstract Figures should not merely recapitulate other figures in the manuscript. Please try to keep the diagram as simple as possible and without superfluous information that may distract from the main conclusion(s). Abstract Figures must be provided by authors no later than the revised manuscript stage and should be uploaded as a separate file during online submission labelled as File Type 'Abstract Figure'. Please also ensure that you include the figure legend in the main article file. All Abstract Figures should be created using BioRender. Authors should use The Journal's premium BioRender account to export high-resolution images. Details on how to use and access the premium account are included as part of this email.

EDITOR COMMENTS

Senior Editor:

Comments to the Author:

Thank you for submitting your study for potential publication in Journal of Physiology. Your manuscript has been reviewed by two expert reviewers, who both felt that it is interesting and that many of the data are useful. There were however some concerns raised that will need to be addressed and a revised manuscript submitted and re-evaluated. All of the points made will need to be addressed. The most important included:

(1) Both reviewers raised concerns that the figures are difficult to understand without more information about recording configuration, labels, etc. As indicated by Reviewer 1, the authors need to pay much more attention overall to the presentation to make it accessible and easily understandable to an audience of non-experts. The impact of the study now is limited by lack of accessibility.

(2) Reviewer 2 raises several caveats about conclusions made, indicating that at least some of the them have clear alternate explanations than the ones provided by the authors. All of these points need to be discussed in the Discussion section. In addition, I would like to see the authors address directly with additional experiments or analysis the possibility that the Arch-induced augmentation of dorsal root responses is due to alkalinization rather than the proposed hyperpolarization. This would appear to be a significant caveat. The reviewer suggests one potential control experiment.

(3) As indicated by Reviewer 2, the authors should strengthen the case that the events in parts i and ii of Fig. 2B are correlated, with a cross-correlation analysis.

(4) As indicated by Reviewer 2, the authors should show that light causes none of the observed effects in wild-type mice.

REFeree COMMENTS

Referee #1:

This paper investigates the role of primary afferents (PA) in the activity of dorsal horn neurons. Using channelrhodopsin or DREADD receptor expressed in PA and recording from neurons in the dorsal horn of the spinal cord, the authors show that PAs coordinate the spontaneous activity of dorsal horn neurons.

The paper is interesting and the methods are solid but the lack of simple illustration to understand what is done in each experiment make the reading of the paper and its access to non-specialists difficult.

Specific points: strong effort to make understandable the set-up and the recording configuration is necessary to catch the attention of physiologists that are not familiar with the spinal cord.

Referee #2:

This is an interesting paper describing the mechanisms behind spontaneous large synchronous afferent activity (termed sDRP), by simultaneously recording from sensory afferents and dorsal horn neurons. Optogenetically depolarizing or hyperpolarizing the afferents decreased or increased afferent activity, respectively, while in contrast silencing afferents with the inhibitory metabotropic construct hm4Di-DREAD inhibited afferent activity and neuron activity. These results are interpreted in the context of known reverberatory spinal circuits by which afferent activity excites interneurons that drive GABAergic neurons that innervate the same afferents, and depolarize them, producing PAD (termed also DRP). PAD in turn alters spike transmission and transmitter release from afferents. This paper provides important new data, but as any good paper, raises more new questions.

Comments:

1) The authors propose that long large depolarizations induced by ChR2 reduce spike transmission by sodium spike inactivation in afferents, leading to reduced synchronous transmitter release that they suggest underly the spontaneous neuronal and sDRP bursts. There is however, no direct evidence shown here for such large effects of sodium inactivation, and previous modelling work suggests that PAD cannot outright block propagation of a spike (Walsmley et al. 1995, Burke et al).

2) The authors further propose that these ChR2 induced depolarizations increase spontaneous transmitter release and lead to increased neuronal firing. This is consistent with their finding that inhibition of all transmitter release with DREADs reduces both sDRP bursts and firing of neurons. However, while Arch induced hyperpolarization of afferents increased sDRP bursts, it did not decrease overall neuron firing, suggesting that transmitter release was not decreased by hyperpolarization, which seems to conflict with the ChR2 results.

3) While these theories are somewhat plausible, the present experiments do not rule out other interpretations. For example, it is probable that Arch and its associate H currents cause an alkalinization of afferents that makes them more excitable and release more transmitter, as recently described in other axons. Specifically, Mahn et al . Nature Neuroscience (2016) show that Arch activation increases terminal Ca due to alkalinization, which increases transmitter release. This can be tested in the current paper by blocking alkalinization with L-lactate in the bathing medium as Mahn did.

4) Furthermore, the ChR2 induced light pulses in Fig 2A produce a synchronous spike at the onset of each pulse which clearly causes a synchronous burst in the neurons. This would be expected to deplete transmitter in the afferents and neurons, and cause a post activation depression lasting many seconds, which would depress spontaneous sDRPs and neuron burst. The burst and associated spiking in the afferent at the onset of a light pulse may not always be seen as it has to propagate antidromically out the dorsal root, but the increase in neuron firing at light onset is always seen, and surely should lead to depression of subsequent activity. These alternate ideas should be discussed. Also, please draw a schematic of the proposed circuits, as it is hard to follow the reasoning without this.

5) In this study light depolarizes all the afferents, including C fibres (fig 1), and root recordings also reflect many populations of afferents, making it hard to tease apart the underlying mechanisms, especially as light was often used a near maximal levels. Perhaps in the future ChR2 expressed in subpopulations of afferents might be useful.

6) To make the figures easier to understand please label the afferents, neurons, light pulses, root stimulation, etc in each panel.

7) The recording in Fig 2Ai looks like it has either a lot of noise, or a slow oscillation. Please show more zoomed in sections over about 100s. Point to the relevant spontaneous activity.

8) In figure 2B, please show cross correlation of i and ii, or at least draw vertical lines at the sDRP bursts, to prove that the neuron activity is lined up with spontaneous PAD events.

9) It would be nice to see records where light was applied in wildtype mice to show it had no effect.

END OF COMMENTS

EDITOR COMMENTS

Senior Editor:

Comments to the Author:

Thank you for submitting your study for potential publication in Journal of Physiology. Your manuscript has been reviewed by two expert reviewers, who both felt that it is interesting and that many of the data are useful. There were however some concerns raised that will need to be addressed and a revised manuscript submitted and re-evaluated. All of the points made will need to be addressed. The most important included:

(1) Both reviewers raised concerns that the figures are difficult to understand without more information about recording configuration, labels, etc. As indicated by Reviewer 1, the authors need to pay much more attention overall to the presentation to make it accessible and easily understandable to an audience of non-experts. The impact of the study now is limited by lack of accessibility.

We have transformed the figures to improve accessibility. Each figure now includes a drawing showing the recording and stimulation conditions, and each trace is specifically labelled in the figure. The figure legends have been also revised. The results section has been modified to make the results easier to understand.

(2) Reviewer 2 raises several caveats about conclusions made, indicating that at least some of the them have clear alternate explanations than the ones provided by the authors. All of these points need to be discussed in the Discussion section. In addition, I would like to see the authors address directly with additional experiments or analysis the possibility that the Arch-induced augmentation of dorsal root responses is due to alkalinization rather than the proposed hyperpolarization. This would appear to be a significant caveat. The reviewer suggests one potential control experiment.

We have revised and completed the discussion section following the concerns of the referee. We have made some experiments with lactate in our ex vivo preparation, but the application of lactate has a profound effect on the parameters analysed, which discourages the use of this approach. We have made some new analysis to check the putative contribution of alkalinisation to the observed results, taking into account its slower time course compared to hyperpolarisation. Considering previous published work with ArchT, we think that alkalinisation could not have a major impact on our results, but it could influence the responses recorded. This issue is discussed in the ms and explained in depth in the response to the referee.

(3) As indicated by Reviewer 2, the authors should strengthen the case that the events in parts i and ii of Fig. 2B are correlated, with a cross-correlation analysis.

We have modified the corresponding figures to include correlograms to illustrate the relationship between the activity in dorsal roots and dorsal horn neurons. We have also used this analysis to include an example of the effect of ArchT increasing the coordination in firing, which we think that is very informative. We have made correlograms for all the recordings included in this paper and we can confirm that coordination is a common feature between dorsal root and dorsal horn neurons in this pool of recordings as well, as we have previously reported (Lucas-Romero et al., 2022 *Front Cell Neurosci* 16:1004956. doi: 10.3389/fncel.2022.1004956).

(4) As indicated by Reviewer 2, the authors should show that light causes none of the observed effects in wild-type mice.

We have included some images in the corresponding figures (current figures 3 and 6) to illustrate the lack of effect of light stimulation at 455 nm and 567 nm in spinal cord preparations extracted from mice that do not express ChR2 or ArchT.

REFEREE COMMENTS

Referee #1:

This paper investigates the role of primary afferents (PA) in the activity of dorsal horn neurons. Using channelrhodopsin or DREADD receptor expressed in PA and recording from neurons in the dorsal horn of the spinal cord, the authors show that PAs coordinate the spontaneous activity of dorsal horn neurons.

The paper is interesting and the methods are solid but the lack of simple illustration to understand what is done in each experiment make the reading of the paper and its access to non-specialists difficult.

Specific points: strong effort to make understandable the set-up and the recording configuration is necessary to catch the attention of physiologists that are not familiar with the spinal cord.

Thanks to the referee to call our attention to this issue, it is very important to make the recording conditions understandable to allow the reader to follow the work properly. We have made great efforts to make the figures more accessible, including drawings of the experimental setup and the recording conditions in all figures. As a consequence the figures and figure legends have changed.

Referee #2:

This is an interesting paper describing the mechanisms behind spontaneous large synchronous afferent activity (termed sDRP), by simultaneously recording from sensory afferents and dorsal horn neurons. Optogenetically depolarizing or hyperpolarizing the afferents decreased or increased afferent activity, respectively, while in contrast silencing afferents with the inhibitory metabotropic construct hM4Di-DREAD inhibited afferent activity and neuron activity. These results are interpreted in the context of known reverberatory spinal circuits by which afferent activity excites interneurons that drive GABAergic neurons that innervate the same afferents, and depolarize them, producing PAD (termed also DRP). PAD in turn alters spike transmission and transmitter release from afferents. This paper provides important new data, but as any good paper, raises more new questions.

The authors would like to thank the referee for revising the manuscript and for his/her suggestions and comments. The discussion now includes a more open framework for the interpretation of the work. The figures have been greatly improved, gaining in clarity and detail.

Comments:

1) The authors propose that long large depolarizations induced by ChR2 reduce spike transmission by sodium spike inactivation in afferents, leading to reduced synchronous transmitter release that they suggest underly the spontaneous neuronal and sDRP bursts. There is however, no direct evidence shown here for such large effects of sodium inactivation, and previous modelling work suggests that PAD cannot outright block propagation of a spike (Walsmley et al. 1995, Burke et al).

As the referee points out, we have no evidence for the putative mechanism(s) responsible for the reduction in sDRP and population burst that we observed. We mentioned sodium channels inactivation because this was suggested by the papers cited in this sentence. Jang et al 2002 wrote "In sensory afferent terminals, the activation of presynaptic GABAA receptors also induces presynaptic inhibition to reduce the electrically stimulated neurotransmitter release resulting from inactivation of Na⁺ channels and/or shunt of the presynaptic membrane by depolarizing the presynaptic terminals" and Zhang & Jackson 1993 that "Depolarization suffices to block action potentials, and the most likely mechanism for this is Na⁺-channel inactivation."

We have deleted this part of the sentence and modified the next one to indicate that depolarisation may interfere with synaptic transmission, which is a more direct and less speculative interpretation of the present results. In light of this and the following comments from the referee, we have added a new paragraph listing some mechanisms that may explain the observations obtained here, all of which are directed to produce different forms of presynaptic inhibition. These mechanisms include actions that can be directly derived from depolarisation, but also other alternatives that can be secondary to the previous activation of synaptic transmission from primary afferents, as the referee indicate. Many mechanisms have been implicated in presynaptic inhibition, but to our knowledge there is no consensus in the literature to which might be the best candidate. A reduction in spike amplitude caused by the inactivation of sodium channels by depolarisation itself or by the shunting effect of opening GABA-A receptors could reduce calcium entry into the presynaptic terminal and subsequent neurotransmitter release. Depolarisation may also inactivate calcium channels, reducing transmitter release. Other work has shown that in proprioceptive afferents, GABA-A receptors may instead have an excitatory effect, favouring the transmission of action potentials in axonal branches, and suggests that the inhibitory effects of GABA may be mediated by activation of GABA-B receptors in these afferents. This latter mechanism, together with others related to neurotransmitter availability, has been implicated in post-activation depression as discussed in point 4. Regardless of the mechanism involved, our results are consistent with a form of inhibition caused by light depolarisation of primary afferents.

2) The authors further propose that these ChR2 induced depolarizations increase spontaneous transmitter release and lead to increased neuronal firing. This is consistent with their finding that inhibition of all transmitter release with DREADs reduces both sDRP bursts and firing of neurons. However, while Arch induced hyperpolarization of afferents increased sDRP bursts, it did not decrease overall neuron firing, suggesting that transmitter release was not decreased by hyperpolarization, which seems to conflict with the ChR2 results.

This is an issue that we have discussed several times in the laboratory. In fact we analysed the data in more detail, trying to find opposite changes between depolarisation and hyperpolarisation. However, the data consistently showed that depolarisation of primary afferents increased spontaneous firing of dorsal horn neurons but decreased evoked responses. Hyperpolarisation did not decrease spontaneous firing, in fact spontaneous firing increased considering the total firing during the 60s light pulse, mostly due to an increase in the firing at population bursts associated to sDRP (now better illustrated in figure 6G and directly stated in the discussion section). Both spontaneous responses increased in number and magnitude during hyperpolarisation. Note that this effect do not implicate a major

change in ongoing firing as occurs with depolarisation. This can be seen in the figure below, which shows a magnification of the recording shown in figure 6B:

Figure shows the effects of a 60s pulse of high-intensity light stimulation (187 mW/cm²; lower panel) on the combined firing frequency of 41 dorsal horn neurons (DHN, upper panel) and dorsal root responses (DR, middle panel). The combined firing of DHN is represented in an enlarged view and truncated at 100 Hz to show the basal firing frequency of DHN. During light stimulation the frequency and amplitude of population bursts increased, but in the periods between bursts (red arrows) the basal firing returns to a similar level than the observed in absence of light (red line).

During hyperpolarisation, the responses evoked by electrical stimulation of the dorsal root did not consistently increase or decrease. As we have tried to explain in the discussion section, we interpret these effects of hyperpolarisation considering that the afferents rest at a membrane potential that prevents spontaneous neurotransmitter release while maximizing synaptic transmission in response to action potentials travelling through the axon when evoked by electrical stimulation. Under these conditions, further hyperpolarisation would not alter continuous spontaneous release, since it is absent, and would not increase action potential-dependent release, since it operated at full capacity at rest. Why does spontaneous activity in dorsal horn neurons associated with sDRP increase with hyperpolarisation? As hyperpolarisation increases the drive for GABA-A-induced depolarisation, this may favour more primary afferents reaching the spike threshold, thus recruiting additional primary afferents (see figure 6H), which would increase the neurotransmitter release and thus activate additional dorsal horn neurons. As

electrical stimulation does activate all the afferents in the absence of light, hyperpolarisation cannot recruit additional afferents.

This view is compatible with our previous work on spontaneous activity, showing that dorsal horn neurons with firing associated with dorsal root activity lose their firing when synaptic transmission is blocked (i.e. dependent on neurotransmitter release; Sci Rep. 2018 Jun 27;8(1):9735. doi: 10.1038/s41598-018-27993-y; Front Cell Neurosci. 2022 Sep 23;16:1004956. doi: 10.3389/fncel.2022.1004956), but other neurons with regular firing appear to fire independently of synaptic transmission (firing by intrinsic mechanisms, no need for continuous neurotransmitter release). This may explain why hyperpolarisation does not affect the firing of regular neurons that may be more implicated in ongoing firing (neurotransmitter-independent), but increases the population burst which consist in firing coordinated by the afferents (neurotransmitter-dependent).

Some sentences have been now included in the results and discussion sections to indicate the difference in the effects of ChR2 and ArchT on spontaneous firing.

3) While these theories are somewhat plausible, the present experiments do not rule out other interpretations. For example, it is probable that Arch and its associate H currents cause an alkalinization of afferents that makes them more excitable and release more transmitter, as recently described in other axons. Specifically, Mahn et al . Nature Neuroscience (2016) show that Arch activation increases terminal Ca due to alkalinization, which increases transmitter release. This can be tested in the current paper by blocking alkalinization with L-lactate in the bathing medium as Mahn did.

Alkalinisation due to archaeorhodopsin (ArchT) activation may be an important issue in the use of this optogenetic tool. There are two nice papers describing alkalinisation in synaptic terminals due to the H⁺ pumping activity of ArchT, and the effects on transmitter release as a consequence of such alkalinisation. Mahn et al 2016 reported calcium accumulation and an increase in spontaneous neurotransmitter release, which increased over the course of 5 minutes of light stimulation. El-Gaby et al 2016 (Cell Rep 16(8):2259-2268. doi: 10.1016/j.celrep.2016.07.057), observed a reduction in field excitatory postsynaptic potentials, indicating a silencing effect on evoked (action potential firing dependent) neurotransmitter release. Mahn et al. also showed an attenuation of evoked postsynaptic currents elicited by electrical stimulation of thalamocortical fibres. However, the inhibitory effect shown by Mahn et al. was observed with short light pulses (200 ms), whereas the reduction seen by El Gaby et al. required more time to occur. In both works, the alkalinisation developed progressively over

minutes in long-lasting light stimulation protocols. Therefore, the effects of alkalinisation appear to have a longer time course than those observed here.

Following the indication of the referee about preventing alkalinisation with L-lactate, we have tried to breed AdvilxArchT mice to do the experiments, but we have had problems with the ArchTxArchT line and we have lost it. We have been trying to obtain animals for the experiments for the last 3 months (even before that for the maintenance of the ArchTxArchT line) and the females are getting pregnant but the litters are not surviving. It is likely that the endogamous breeding that has been carried out in our facility for several years has fixed any genetic defect that is detrimental to breeding or survival. We considered the possibility of re-acquiring the line from the Jackson laboratories, but in the meantime, we have done some experiments on wild type animals which discourage the validity of this experiment under our experimental conditions.

We have made three experiments to test the effects of 50 mM of sodium lactate on spinal cord responses recorded from wild-type mice. During the application of lactate to the preparation, we have observed a strong excitatory effect on the spontaneous activity of dorsal horn neurons, which developed a constant and disorganised firing. In addition, sDRP were completely abolished. Lactate has a direct effect on the parameters measured in the present work, making the proposed study impossible. Please see an example below. Taking as reference the work of Mahn et al, 2016, we replaced 50 mM of NaCl with 50 mM Na-lactate, to maintain the molarity of the ACSF. This resulted in a 50mM decrease in the concentration of chloride ions. It is likely that the reduction in chloride levels to incorporate lactate makes the spinal circuits more excitable, leading to dysregulated firing. The abolition of sDRP could also be related to the uncoordinated activity of dorsal horn neurons and/or the alteration of chloride gradients. Both effects strongly resemble the effect observed depolarising the afferents by ChR2 activation. Mahn et al used this manoeuvre in culture, but not in their slice preparation, perhaps for this reason. In a complex model, any alteration in the conditions may have a stronger impact in the functioning of the circuits.

A**L-Lactate 50 mM****B****C**
Effects of L-lactate on the spontaneous activity of dorsal horn neurons (DHN) and primary afferents contained in a dorsal root (DR). Example of a continuous recording showing the combined firing of 24 dorsal horn neurons recorded with 32-channels MEA (upper traces) and the recording from a dorsal root (lower traces). Panel A show the whole experiment including the signals before and during the application of the modified ACSF with the addition of 50mM sodium L-lactate and the reduction of 50 mM NaCl (blue line). The dorsal root recording has been processed to remove the strong drift caused by the application of L-Lactate. When the lactate-ACSF reached the recording chamber, the signal dropped by several millivolts, probably due to the action of reducing chloride ions on the Ag-AgCl pellet used as reference electrode. An expanded view is shown below for the areas marked with 1 and 2 in A. B shows an expanded fragment of the recording in control conditions, showing sDRP and population bursts. C shows an equivalent fragment during L-lactate application (approximately 14 minutes after application onset). The sDRP were abolished and the firing of DHN increased, but the coordinated events were lost.

We have performed a search on papers citing Mahn et al 2016 to find others work using L-lactate to avoid ArchT-induced alkalinisation, without success. Some of these papers consider that during light stimulation using pulses ≤ 1 minute the main effect of ArchT is silencing due to the hyperpolarisation, and excitatory effects are

not reported with this stimulus duration. We defined our protocol to have a time window that allowed the analysis of a sufficient amount of sDRP and associated activity, and one minute provided enough time. Several other works have used stimulation times of around 1 minute and consistently show inhibitory effects (Asok et al., 2018 *Mol Psychiatry*. 23(4):914-922. doi: 10.1038/mp.2017.79; Falgairolle et al., 2017 *Elife* 6:e26622. doi: 10.7554/eLife.26622; Garcia-Garcia et al., 2018 *Mol Psychiatry*. 23(10):1990-1997. doi: 10.1038/mp.2017.165; Kaur et al., 2017 *Neuron*. 96(5):1153-1167.e5. doi: 10.1016/j.neuron.2017.10.009; Wolfman et al., 2018 *Nat Commun* 9(1):2710. doi: 10.1038/s41467-018-04654-2; Thankachan et al., 2019 *Sci Rep*. 9(1):3607. doi: 10.1038/s41598-019-40398-9; Twining et al., 2020 *J Neurosci*. 40(16):3217-3230. doi: 10.1523/JNEUROSCI.1453-19.2020; Mahadevia et al., 2021 *Nat Commun*. 12(1):6796. doi: 10.1038/s41467-021-27092-z.)

As mentioned above, alkalinisation seems to develop slowly during light application. To test whether the effects we observed with ArchT stimulation were progressive, we have analysed in detail the firing of dorsal horn neurons for the maximum light intensity used. We have analysed separately the firing associated and not associated with sDRP. If activation of archT leads to alkalinisation and an increase in transmitter release, a progressive increase in ongoing firing would be expected. In all the recordings, the firing associated to sDRP and the firing between sDRP (ongoing) is variable for each event, but it did not show a tendency to increase during the 60s light stimulation. In fact, it was observed more often a tendency to slightly decrease. The graph below shows a quantification from a representative experiment.

Graph summarising the frequency of action potentials fired by 54 dorsal horn neurons in association with sDRP (between 200 ms before and 50 ms after the peak of the sDRP, in black) and in the time window between sDRP (variable duration for each point, red). Each point represents the firing associated with a single sDRP that peaked at the time indicated in the X-axis (light onset at 0s) or the time in the middle between two consecutive sDRP for non-associated firing. Note that firing frequency is higher in association with sDRP (different scale between Y-axes).

We have included some sentences in the discussion section indicating that ArchT activation may cause alkalinisation in the presynaptic terminal, which could lead to alterations in synaptic transmission. Although the main effects shown can be explained by hyperpolarisation, changes in synaptic terminal excitability may contribute to the observed results. Optogenetic manipulation may alter the function of complex circuits, complicating the interpretation of results as recently reported (Watkins de Jong et al., 2023 *Curr Biol* 33(9):1689-1703.e5. doi: 10.1016/j.cub.2023.03.032), and we cannot definitively exclude a role of alkalinisation.

4) Furthermore, the Chr2 induced light pulses in Fig 2A produce a synchronous spike at the onset of each pulse which clearly causes a synchronous burst in the neurons. This would be expected to deplete transmitter in the afferents and neurons, and cause a post activation depression lasting many seconds, which would depress spontaneous sDRPs and neuron burst. The burst and associated spiking in the afferent at the onset of a light pulse may not always be seen as it has to propagate antidromically out the dorsal root, but the increase in neuron firing at light onset is always seen, and surely should lead to depression of subsequent activity. These alternate ideas should be discussed. Also, please draw a schematic of the proposed circuits, as it is hard to follow the reasoning without this.

As the referee point out, light stimulation evokes firing in primary afferents. This is evident in vivo by the generation of reflex responses and also ex vivo in both peripheral nerve and dorsal root recordings (current figures 1 and 2, respectively). Firing in primary afferents induced by light stimulation, like electrical stimulation, results in the release of neurotransmitters that elicit synaptic responses in dorsal horn neurons and afferents. This activity may alter the function of spinal circuits and influence their subsequent responses. As we have explained in the response to comment 1, we have added new sentences in the discussion section to indicate that multiple mechanism can be involved in the observed effects. Our approach is adequate to investigate the general function of the system in a native conformation, but does not allow identification of the mechanisms responsible using extracellular recordings.

Although previous activation of primary afferent can alter neurotransmitter release, the abolition of sDRP and population burst is maintained during the 60 seconds pulse, which is probably too long for postsynaptic depression in primary afferents (J Physiol. 2023 May;601(10):1925-1956. doi: 10.1113/JP283706.). In addition, both electrical activation of primary afferents and stimulation with brief light pulses also produced synaptic activation, but they did not induce such long-lasting inhibition in sDRP and population bursts (see example below; panels A and B).

Another argument in this direction comes from ramp depolarisations. We have tested this protocol in isolated nerves and confirmed that slowly increasing the intensity of light stimulation (from 0 to 42 mW/cm² during 20 seconds) does not produce any observable response in primary afferents (recorded in vitro in saphenous and sciatic nerves, not shown). This is also true for recordings from dorsal roots, where no volleys can be observed with this procedure, as shown in the present manuscript (current figure 2F). This ramp stimulation produced the same long lasting abolition of sDRP and population burst as pulse stimulation (see panel C below), suggesting that the maintained depolarisation itself may play some role in the observed effects. In these circumstances, we still think that a mechanism of inhibition of primary afferents more directly related to depolarisation is probably contributing to some extent to the observed effects. Anyway, the mechanistic explanation does not alter the main conclusions of this work, which are that primary afferents coordinate the occurrence of spontaneous activity in the spinal cord and that altering their capacity for synaptic transmission impedes coordination.

Image in A shows the responses of dorsal horn neurons (DHN) and a dorsal root (DR) to a single high-intensity electrical stimulus applied to an adjacent dorsal root (black triangle). B shows the same responses after a light stimulus of 50 ms at an irradiance of 42 mW/cm² (arrow). Note that a few seconds after the primary afferent activation for both types of stimuli, the sDRP (lower trace) and the coordinated firing of DHN in the form of population bursts (upper trace) are again visible. In contrast a slow depolarisation induced by a progressive increase in the light stimulus intensity (from 0 to 42 mW/cm²) over 20 seconds maintained a continuous

depolarisation, abolishing sDRP but also the occurrence of population bursts. All images were taken from the same experiment, but please note that the amplification of the dorsal root signal and the time shown are different in each image.

We have been studying spontaneous activity in dorsal horn circuits for several years and we have defined a simplified model of the operating circuit, which can be seen below (Lucas-Romero et al., 2022 Front Cell Neurosci 16:1004956. doi: 10.3389/fncel.2022.1004956; a modified version can also be found in Lucas-Romero et al., 2024 Pflugers Arch 476(8):1171-1186. doi: 10.1007/s00424-024-02971-8)

Primary afferents transmit sensory information from peripheral receptors into the spinal cord, where they make contact with postsynaptic neurons. In addition, primary afferents receive presynaptic contacts from dorsal horn neurons (including GABAergic neurons) that regulate the transmission of incoming information. Both primary afferents and dorsal horn neurons show spontaneous activity under ex vivo conditions, and this activity is highly coordinated. In the present work, we show that depolarisation of primary afferents abolished activity in the dorsal root and coordination in dorsal horn neurons, whereas hyperpolarisation increased responses in the dorsal root and enhanced coordinated activity. This supports the idea that rhythmic activity received by primary afferents is spread to postsynaptic neurons, generating synchronous spontaneous events.

5) In this study light depolarizes all the afferents, including C fibres (fig 1), and root recordings also reflect many populations of afferents, making it hard to tease apart the underlying mechanisms, especially as light was often used a near maximal levels. Perhaps in the future ChR2 expressed in subpopulations of afferents might be useful.

With this approach, we wanted to investigate the general role of primary afferents and therefore chose this model for the studies. We agree with the referee that the use of a model that include all the primary afferents makes impossible to attribute the observed effects to specific subpopulations of afferents, and future studies will be needed to define the subpopulations involved. We have added a new sentence to the first paragraph in the discussion section to call the attention to this issue. A more specific model will allow direct manipulation of a defined population, but as the referee pointed out, recording of the whole root does not allow identification of specific types of afferents.

6) To make the figures easier to understand please label the afferents, neurons, light pulses, root stimulation, etc in each panel.

Following the indication of both referees, we have made major changes to the figures to improve the clarity. We have included drawings of the recording conditions for each experiment and labelled the figures in a more accessible form. As a consequence, the figures and figure legends have changed.

7) The recording in Fig 2Ai looks like it has either a lot of noise, or a slow oscillation. Please show more zoomed in sections over about 100s. Point to the relevant spontaneous activity.

The image in former fig 2Ai (now 3B and labelled as DHN) represents the combined responses of the entire population of dorsal horn neurons recorded simultaneously with multielectrode arrays, and is therefore not a direct recording. This extracellular approach allows recording the firing of action potentials from a large amount of neurons with good temporal resolution. Once the occurrence of action potential firing in each identified neuron had been isolated, the firing times of all recorded neurons were combined. This combined firing is shown as the firing frequency for the entire recorded population (in this image in figure 3 we were able to isolate 70 different neurons). Since there are neurons with different firing patterns of spontaneous activity, the combination results in a complex sequence. This allows the analysis of coordinated activity in the form of population bursts, but there are also small oscillations in the firing frequency, as the referee noted.

In the new figure 3, the sections now labelled C and E show two expanded fragments of the experiment shown in B (the expanded area is now indicated by the numbers 1 and 2 in figure 3B). As can be seen, the oscillation is more pronounced

in fig 2E due to the increased firing of dorsal horn neurons during depolarisation (note that the basal level of firing frequency is also increased compared to the values in the absence of light, fig 3C). This oscillation probably reflects asynchrony in DHN firing. We have constructed the figures with 20 seconds time range to improve visualisation, but please find below enlarged sections of 100s from the same areas as requested (note that as light stimulus lasted 60s, the image contains areas before, during and after light).

A

Dorsal horn neurons (DHN)

200 Hz |

200 μ V |
10 s

Dorsal root (DR)

B

DHN

DR

455nm; 42 mW/cm²

The experiment shown in figure 3B (figure 2A in the previous version) is shown in an extended time base to observe 100s fragments in control and during light application to activate ChR2. The image in A shows the spontaneous responses of 70 dorsal horn neurons (DHN) and one dorsal root (DR) in absence of any type of stimulation. B shows the same preparation in the next time section, during 60 s light stimulation at 42 mW/cm² (blue line).

8) In figure 2B, please show cross correlation of i and ii, or at least draw vertical lines at the sDRP bursts, to prove that the neuron activity is lined up with spontaneous PAD events.

We have now plotted some lines in figure 3C (former figure 2B) to show the temporal correspondence between sDRP and dorsal horn neuron firing, and also a correlogram as figure 3G to show the relationship between the two signals. A clear peak can be seen with this analysis.

We have also included an additional correlogram in figure 6G showing the relationship between the two signals and the effect of hyperpolarisation of the primary afferents on this correlation.

The procedure to obtain this analysis is now explained in the methods section.

9) It would be nice to see records where light was applied in wildtype mice to show it had no effect.

We have included new images in the corresponding figures (current figures 3 and 6) showing examples of two different experiments in the spinal cord of mice that do not express channelrhodopsin-2 or archaerhodopsin-3. Following exactly the same protocol of light stimulation, there were no light effects.

Dear Dr Rivera-Arconada,

Re: JP-RP-2025-287970R1 "Central terminals of primary afferents coordinate the spontaneous activity of dorsal horn neurons." by Javier Lucas-Romero, Jose Antonio Lopez-Garcia, and Ivan Rivera-Arconada

We are pleased to tell you that your paper has been accepted for publication in The Journal of Physiology.

Yours sincerely,

Nathan Schoppa
Senior Editor
The Journal of Physiology

If you would like to receive our 'Research Roundup', a monthly newsletter highlighting the cutting-edge research published in The Physiological Society's family of journals (The Journal of Physiology, Experimental Physiology, Physiological Reports, The Journal of Nutritional Physiology and The Journal of Precision Medicine: Health and Disease), please click this link, fill in your name and email address and select 'Research Roundup':
<https://www.physoc.org/journals-and-media/membernews>

- **TRANSPARENT PEER REVIEW POLICY:** To improve the transparency of its peer review process, The Journal of Physiology publishes online as supporting information the peer review history of all articles accepted for publication. Readers will have access to decision letters, including Editors' comments and referee reports, for each version of the manuscript as well as any author responses to peer review comments. Referees can decide whether or not they wish to be named on the peer review history document.
- You can help your research get the attention it deserves! Check out Wiley's free Promotion Guide for best-practice recommendations for promoting your work at: www.wileyauthors.com/eeo/guide. You can learn more about Wiley Editing Services which offers professional video, design, and writing services to create shareable video abstracts, infographics, conference posters, lay summaries, and research news stories for your research at: www.wileyauthors.com/eeo/promotion.
- **IMPORTANT NOTICE ABOUT OPEN ACCESS:** To assist authors whose funding agencies mandate public access to published research findings sooner than 12 months after publication, The Journal of Physiology allows authors to pay an Open Access (OA) fee to have their papers made freely available immediately on publication.

EDITOR COMMENTS

Reviewing Editor:

Senior Editor:

Thank you for your thoroughly revised manuscript. The two expert referees who reviewed your original submission are satisfied with the changes that you have made, including those that have made the manuscript more accessible and more

complete in terms of discussion of caveats/alternative interpretations. Congratulations! The work is now acceptable for publication.

REFEREE COMMENTS

Referee #1:

The manuscript has been adequately revised. In particular, the paper is now clarified. I have no further comments.

Referee #2:

Nice work on revisions. I have no further comments.